# Untangling Component Imbalance in Hybrid Linear Attention Conversion Methods

## Abstract

Transformers' quadratic computational complexity limits their scalability despite remarkable performance. While linear attention reduces this to linear complexity, pre-training such models from scratch remains, in most cases, prohibitively expensive. Recent post-training linearisation methods convert pre-trained Transformers to linear models efficiently, often using hybrid approaches that combine linear attention with sliding-window softmax. We identify a critical flaw: existing hybrid methods inadvertently bypass the linear component, relying almost entirely on the sliding-window. Component-level diagnostics reveal this previously undetected behaviour stems from overlooked evaluation practices on common-sense benchmarks. We propose three solutions to ensure balanced component usage: (i) inference-time hybridisation of linear-only conversions with sliding-window softmax; (ii) HedgeCATs, combining attention-weight transfer with targeted LoRA fine-tuning; and (iii) Scheduled Sliding-window Dropout (SSD), which stochastically suppresses the softmax branch during training to prevent component collapse. Our methods maintain computational efficiency while recovering most base model performance and ensuring genuine linear attention adoption, restoring the validity of performance attributions in hybrid conversions.

## 1 Introduction

Transformers (Vaswani et al., 2017) have delivered state-of-the-art results across language, vision, and multimodal tasks, yet their quadratic attention cost in sequence length remains a central bottleneck for long-context inference and training. Linear attention (LA) (Katharopoulos et al., 2020), offers a compelling alternative by replacing the softmax kernel with linear feature maps that enable associative, streaming updates of a fixed-size recurrent state (Choromanski et al., 2021; Duman Keles et al., 2023; Banerjee et al., 2020; Peng et al., 2021; Qin et al., 2022). In principle, this reduces the asymptotic complexity of both memory and compute compared to softmax attention. In practice, however, fully pre-training LA models is costly (Liu et al., 2020), and performance often lags behind quadratic baselines trained with similar budgets due to limitations in representational complexity (Zhang et al., 2024b; Mercat et al., 2024).

A growing body of work focuses on circumventing the high cost of pre-training linear models via post-training linearisation (Zhang et al., 2024b;a; Lan et al., 2025; Mercat et al., 2024): converting a pre-trained quadratic Transformer into a fully linear or hybrid linear-softmax model. This approach amortises most of the cost into the pre-trained base model and performs a light "swap + adaptation" stage, in which the softmax kernel is replaced with a learnable linear kernel, followed by additional pre-training and/or supervised fine-tuning to recover performance. Such methods have been reported to require on the order of $0.02\%$ (or less) of the data used to train the base model to recover performance. Existing conversion methods typically differ in (i) the LA formulation, (ii) whether and how the original weights are fine-tuned, and (iii) whether sliding-window softmax attention (SWA) (Beltagy et al., 2020; Zaheer et al., 2020) is retained alongside the linear path, yielding hybrid architectures.

Hybrid conversions are attractive as they pragmatically combine the representational capacity of SWA with the computational efficiency of LA. This makes them an attractive direction for long-context tasks. Yet, as we show, these reported gains can mask a critical failure mode: the model may lean almost entirely on the SWA path while effectively ignoring the linear component, which, by itself,

performs no better than removing attention entirely. This creates misleading performance attribution: the hybrid is credited for "using LA" when, in reality, the model fails to learn a meaningful linear kernel and simply biases and adapts itself towards SWA entirely during the fine-tuning stage. The field lacks standard diagnostics to quantify each component's contribution, so such imbalances can remain hidden behind aggregate metrics.

**Contributions** (1) We identify and characterise a systematic issue in current hybrid attention conversions whereby models learn to ignore their LA component and over-rely on their SWA one, leading to misleading attribution of hybrid performance. (2) We provide component-level diagnostic that make this imbalance visible and reproducible across popular pre-trained base models on standard common-sense benchmarks. (3) We introduce three practical remedies: a zero-shot inference-time hybrid; HedgeCATs, which combines HedgeHog-style attention-weight transfer with brief LoRA fine-tuning; and Scheduled Sliding-window Dropout (SSD) to prevent component imbalance during training. We show that our proposed strategies recover most base-model performance while ensuring genuine use of the LA pathway, restoring attributional validity without sacrificing computational efficiency.

## 2 Background & Related Works

### 2.1 Linear Attention

Let $\mathbf{X} \in \mathbb{R}^{T \times d_{\text{model}}}$ be a sequence of length $T$ with projections $\mathbf{Q} = \mathbf{X}\mathbf{W}_Q, \mathbf{K} = \mathbf{X}\mathbf{W}_K, \mathbf{V} = \mathbf{X}\mathbf{W}_V$, where $\mathbf{Q}, \mathbf{K} \in \mathbb{R}^{T \times d}$ and $\mathbf{V} \in \mathbb{R}^{T \times d_v}$. Standard softmax attention (Vaswani et al., 2017) computes

$$\mathbf{O} = \text{softmax}\left(\frac{\mathbf{Q}\mathbf{K}^\top}{\sqrt{d}}\right)\mathbf{V} \tag{1}$$

incurring $O(T^2)$ complexity for the $T \times T$ similarity matrix. Using a kernel $\kappa(\mathbf{q}, \mathbf{k})$ such that $\kappa(\mathbf{q}, \mathbf{k}) \approx \phi(\mathbf{q})^\top \phi(\mathbf{k})$ for a non-negative feature map $\phi : \mathbb{R}^d \to \mathbb{R}^{d'}$ as proposed by Katharopoulos et al. (2020), we can avoid forming pairwise similarities by introducing global summaries:

$$\mathbf{o}_t = \frac{\sum_{i=1}^T \kappa(\mathbf{q}_t, \mathbf{k}_i)\, \mathbf{v}_i}{\sum_{i=1}^T \kappa(\mathbf{q}_t, \mathbf{k}_i)} = \frac{\phi(\mathbf{q}_t)^\top \mathbf{S}}{\phi(\mathbf{q}_t)^\top \mathbf{z}}, \quad \mathbf{S} = \sum_{i=1}^T \phi(\mathbf{k}_i)\mathbf{v}_i^\top, \ \ \mathbf{z} = \sum_{i=1}^T \phi(\mathbf{k}_i). \tag{2}$$

These summaries can be accumulated in a single pass, so no $T \times T$ score matrix is materialised. This yields $O(Tdd')$ time and $O(dd')$ memory. Proposed $\phi$ in the literature include non-negative element-wise maps (such as $1 + \text{ELU}(\mathbf{x})$ (Katharopoulos et al., 2020) and $\text{ReLU}$ (Kasai et al., 2021)), random feature maps (Choromanski et al., 2021), and exponential function approximations via low-order Taylor expansions (Duman Keles et al., 2023; Banerjee et al., 2020).

### 2.2 Hybrid Attention

Hybrid attention mechanisms combine softmax attention with LA through various architectural approaches. Some methods interleave full softmax attention layers with LA layers (Lieber et al., 2024; Dong et al., 2024), while others employ SWA combined with LA, either in interleaved layers (Ren et al., 2024) or integrated within the same Transformer block (Beltagy et al., 2020; Zhang et al., 2024a; Lan et al., 2025; Irie et al., 2025; Munkhdalai et al., 2024). The adoption of SWA offers the key advantage of preserving linear complexity throughout the entire model. Following other hybrid conversion approaches (Zhang et al., 2024a; Lan et al., 2025; Irie et al., 2025), our work focuses on combining SWA with LA within Transformer blocks to approximate full softmax attention. These methods typically employ a scaled linear combination of the two attention outputs, utilising learned (Munkhdalai et al., 2024; Zhang et al., 2024a), fixed (Lan et al., 2025), or data-dependent scaling factors (Behrouz et al., 2024; Irie et al., 2025). This can be summarised by scalar or vector mixing terms $a, b$, such that the hybrid attention output is given by:

$$\mathbf{ATTN}(Q, K, V, a, b) = a \odot \mathbf{SWA}(Q, K, V) + b \odot \mathbf{LA}(Q, K, V) \tag{3}$$

## 2.3 Linearising Pre-Trained Transformers

Although numerous linear (Schlag et al., 2021; Gu & Dao, 2023; Yang et al., 2023; 2024b; Peng et al., 2025) and hybrid (Beltagy et al., 2020; Zhu et al., 2021; Lieber et al., 2024; Yang et al., 2024b; Behrouz et al., 2024) Transformers have been developed, the majority are trained from scratch. The prohibitive cost of pre-training constrains most of these approaches to small model sizes (typically $< 1B$ parameters) and makes them expensive to reproduce or scale when checkpoints are unavailable, directly undermining the central promise of linear Transformers: computational efficiency. This limitation renders post-training conversion methods particularly appealing, as they offer the potential for high performance recovery in large-scale Transformers at a fraction of pre-training costs (typically $< 1\%$).

While LA is theoretically equivalent to softmax-based self-attention under feature map $\phi$ and kernel $\phi(\boldsymbol{q}_t)^\top \phi(\boldsymbol{k}_i) = \exp\left((\boldsymbol{q}_t^\top \boldsymbol{k}_i) \cdot D^{-1/2}\right)$ (Katharopoulos et al., 2020), such a (finite-dimensional) feature map does not currently exist. As such, converting a pre-trained Transformer to use LA requires some adjustments and fine-tuning to make up for the change in attention weights.

Proposed kernels are designed to ensure positive attention weights via non-linear activation functions (Katharopoulos et al., 2020; Kasai et al., 2021; Mercat et al., 2024; Zhang et al., 2024b;a). However, these non-negative activation functions run the risk of suppressing any negative signals and may unnecessarily constrain the learned mappings. To this end, Zhang et al. (2024b) and Zhang et al. (2024a) concatenate the negative mapping to the output along the head dimension, applying their respective non-linear activation function $\sigma$ to each separately:

$$\phi(\boldsymbol{x}) = [\sigma(\boldsymbol{W}_\phi^\top \boldsymbol{x} + \boldsymbol{b}) \oplus \sigma(-\boldsymbol{W}_\phi^\top \boldsymbol{x} - \boldsymbol{b})] \tag{4}$$

Zhang et al. (2024b) showed that the softmax function's unique spikiness and monotonicity with respect to the Query-Key dot-product are hard to match when using previously proposed candidates for $\phi$. They therefore learn an exponential feature map (Eq. 4 with $\sigma = \exp(\cdot)$), and train it via an attention-weights transfer objective that minimises cross-entropy between softmax attention weights and linear weights. This "weights-to-weights" stage is followed by a fine-tuning stage of the original model weights which they claim makes up for any approximation errors. LoLCATs (Zhang et al., 2024a) then later explores general conversion methods using LoRA (Hu et al., 2022) fine-tuning as well as hybrid attention methods. In parallel, SUPRA (Mercat et al., 2024) follows T2R (Kasai et al., 2021) in adopting a ReLU-activated feature map with standard language-model fine-tuning rather than an explicit attention-transfer loss. This keeps training simple since no new parameters are introduced.

Other conversion methods (Mao, 2022; Lan et al., 2025) focus on converting pre-trained Transformers to gated linear/recurrent blocks (GLA-style). Both these methods repurpose the attention block into a gated linear update; Liger (Lan et al., 2025) differs in explicitly retaining a local softmax branch. A summary of different linear kernels and transfer objectives for various conversion methods can be found in Table 5.

## 2.4 Training Interventions to Tackle Component Collapse

As explored in this paper, hybrid attention conversion models can learn to ignore the linear path and rely solely on SWA. Related work in other settings tackles analogous "path collapse" with structured dropout (Srivastava et al., 2014): dropping whole substructures during training so models learn to balance all their trained components. In Transformers, DropHead (Zhou et al., 2020) targets multi-head attention (Cordonnier et al., 2021) directly by stochastically dropping whole heads with a scheduled rate to prevent a few heads from monopolising computation. Other works explore similar ideas but applied to different subcomponents such as Transformer layers (Fan et al., 2020), experts (Chen et al., 2023) in Mixture-of-Experts models, and even incoming keys (Li et al., 2023).

## 3 Identifying the Issues within Hybrid Conversion Methods

In this section, we outline some issues we have found to occur in conversion methods which make use of a hybrid attention-based training objective. We start by re-implementing and ablating the LoLCATs framework, which is considered to be the state-of-the-art (SOTA) method for converting pre-trained

Transformers to use hybrid attention, as well as repeating such analysis in their own codebase and checkpoints. We complete these findings with a component-wise investigation of similar methods, ablating key components with those used in similar SOTA LA-only methods, one at a time, in order to identify the ones responsible for the issues observed in hybrid methods. Further ablations and complementary analyses can be found in Appendix A.6.

## 3.1 EXPERIMENTAL SET-UP

**Feature Map** $\Phi$    We adopt a learned feature map (Mercat et al., 2024; Zhang et al., 2024b;a), with $W_\phi \in \mathbb{R}^{h_d \times \frac{h_d}{2}}$ and a softmax activation function (ie. Eq 4 with $\sigma = \text{softmax}(\cdot)$), and apply the RoPE embeddings to queries and keys prior to applying $\phi$, as motivated by Zhang et al. (2024a).

**Combining Sliding-Window and Linear Attention**    For our experiments, we implement Eq. 3 with a simple choice of $a = g$, $b = 1 - g$, with fixed $g = \frac{1}{2}\mathbf{1}$, as used in Lan et al. (2025). We avoid learned or dynamic mixing terms, as well as a shared denominator, to ensure that LA and SWA are used equally in the hybrid attention outputs. The LA component only operates on tokens outside the sliding-window, which has size $64$.

**Models and Datasets**    Our experiments are focused around converting and evaluating to three popular, pre-trained Transformers: Mistral-7B-Instruct-v0.1, Llama-3-8B-Instruct, and Llama-3.1-8B-Instruct (more models, sizes, and checkpoints in Appendix A.6). Attention transfer and fine-tuning are carried out on truncated samples of 1024 tokens from the FineWeb-Edu dataset (Penedo et al., 2024). For evaluation, we follow other conversion methods (Zhang et al., 2024a; Lan et al., 2025) and report performance on popular LM-Eval tasks, all zero-shot: PIQA, ARC-Easy, ARC-Challenge (acc norm), HellaSwag (acc norm), WinoGrande, and MMLU. In all results, any average performance shown is calculated across all six tasks.

**Training & LoRA Parameters**    We follow Zhang et al. (2024a)'s exact fine-tuning settings, applying LoRA to the query, key, value, and output projection matrices ($W_q$, $W_k$, $W_v$, $W_o$), with rank $r = 8$, $\alpha = 16$. We use an AdamW (Loshchilov & Hutter, 2017) optimiser with learning rate $1e-4$ ($1e-2$ during attention transfer), and a reduce-on-plateau scheduler. We train for 1 epoch for attention transfer, and 1-5 epochs during fine-tuning, using an effective (accumulated) batch size of 64 and approximately 25M tokens per epoch.

## 3.2 ABLATION STUDY OF LOLCATS HYBRID ATTENTION MODULES AT INFERENCE TIME

In this section, we explore the contribution of each attention module to performance for models trained using LoLCATs' proposed hybrid conversion methodology (Zhang et al., 2024a). We run four ablations at inference time: (i) SWA-only, removing the LA component; (ii) LA-only, removing the SWA component; (iii) attention sinks (Xiao et al., 2023) only, suppressing both SWA and LA and only passing the first $8$ values through softmax attention; and (iv) no attention, where we return an all-zeros attention output, removing any contribution from the attention mechanism. Additionally, we run the same ablations using the provided LoLCATs-trained checkpoints for Llama-3.1-8B using the LoLCATs codebase. Results are shown in Table 1 and per-task results with standard errors can be found in Appendix A.5.1.

Together, the ablations show that almost all the performance attribution sits within the SWA component. Using SWA-only in the resulting models achieves very similar accuracy, and, in the case of both Mistral and the LoLCATs-trained Llama-3.1-8B [1], either an insignificant decrease or a significant improvement in performance. By contrast, LA-only, attention sinks only, and no attention all collapse to roughly the same low performance. The results expose a clear SWA-LA imbalance in current hybrid attention conversion methods that leads to LA contributing little to downstream accuracy or even being detrimental. Similar findings are reported in Lan et al. (2025) (see their Table 6) where they find that SWA-only and GLA+SWA lead to very similar performance while GLA-only leads to a dramatic decrease in performance.

---

[1] https://huggingface.co/hazyresearch/lolcats-Llama-3.1-8b-distill https://huggingface.co/hazyresearch/lolcats-Llama-3.1-8b-ft-lora

| Active Attn Modules | PIQA | ARC-E | ARC-C | HellaSwag | WG | MMLU | AVG | Rec. Perf |
|---|---|---|---|---|---|---|---|---|
| *Mistral-7B-Instruct* | 79.27 | 80.01 | 52.22 | 74.60 | 69.93 | 53.51 | 68.26 | 100.00 |
| SWA + Linear | **78.24** | **79.59** | **50.42** | 71.19 | 68.09 | 45.71 | 65.54 ± 0.07 | 96.02 ± 0.10 |
| SWA only | 78.02 | 79.46 | 49.77 | 68.28 | **68.51** | 46.90 | 65.16 ± 0.21 | 95.46 ± 0.30 |
| Linear only | 53.75 | 29.73 | 25.08 | 27.21 | 50.54 | 23.12 | 34.91 ± 0.07 | 51.14 ± 0.10 |
| SWA + Attn sinks | 78.17 | 79.50 | 50.17 | **73.37** | 68.38 | **48.66** | **66.38** ± 0.08 | **97.24** ± 0.12 |
| Attn sinks only | 66.90 | 64.02 | 39.56 | 32.00 | 61.49 | 44.81 | 51.46 ± 0.23 | 75.40 ± 0.34 |
| No Attention | 53.92 | 25.63 | 24.66 | 25.99 | 50.67 | 25.51 | 34.40 ± 0.00 | 50.39 ± 0.00 |
| *Llama-3-8B-Instruct* | 78.13 | 81.69 | 56.66 | 75.94 | 71.67 | 63.85 | 71.32 | 100.00 |
| SWA + Linear | 78.02 | 80.15 | 53.78 | 71.93 | 71.85 | 48.78 | 67.42 ± 0.07 | 94.53 ± 0.09 |
| SWA only | 78.09 | 80.00 | 54.01 | 64.93 | 71.88 | 47.05 | 65.99 ± 0.05 | 92.52 ± 0.06 |
| Linear only | 52.90 | 25.71 | 26.02 | 26.47 | 48.72 | 22.96 | 33.80 ± 0.14 | 47.39 ± 0.20 |
| SWA + Attn sinks | **78.33** | **80.68** | **54.52** | **76.55** | **71.95** | **58.00** | **70.01** ± 0.04 | **98.15** ± 0.06 |
| Attn sinks only | 55.40 | 30.40 | 22.61 | 28.99 | 50.78 | 22.95 | 35.19 ± 0.18 | 49.34 ± 0.25 |
| No Attention | 55.17 | 26.73 | 22.87 | 26.13 | 51.14 | 22.95 | 34.17 ± 0.00 | 47.91 ± 0.00 |
| *Llama-3.1-8B (LoLCATs ckpt)* | 80.14 | 81.82 | 55.20 | 79.14 | 73.72 | 68.05 | 73.01 | 100.00 |
| SWA + Linear | 81.18 | 82.37 | 54.78 | 79.16 | 70.09 | 58.89 | 71.08 | 97.35 |
| SWA only | 81.56 | 82.37 | 55.29 | 79.76 | **74.11** | 55.63 | 71.45 | 97.87 |
| Linear only | 51.52 | 25.00 | 25.51 | 26.37 | 52.25 | 23.09 | 33.96 | 46.51 |
| SWA + Attn sinks | **81.66** | **82.45** | **55.38** | **79.85** | **74.11** | **61.21** | **72.44** | **99.22** |
| Attn sinks only | 61.37 | 41.62 | 21.93 | 29.85 | 49.80 | 23.12 | 37.95 | 51.98 |
| No Attention | 54.62 | 26.68 | 24.40 | 25.88 | 48.86 | 23.12 | 33.93 | 46.47 |

Table 1: Measuring the effect of attention components on benchmark accuracy for models trained with LoLCATs hybrid conversion. Standard errors with $N = 3$ are shown for the main ablations.

### 3.3 Reverting Back to Linear Attention-only

In this section, we re-examine the HedgeHog (Zhang et al., 2024b) pre-trained conversion method. HedgeHog has been shown to work with both, task-specific full-parameter fine-tuning and general pre-trained converion using LoRA, and may be considered the SOTA conversion method for LA-only. Hence, we simply seek to investigate whether their results extend to more recent models used by LoLCATs, or whether this difference may be responsible for the gap in LA-only performance between HedgeHog and LoLCATs. More specifically, we evaluate models converted using HedgeHog's core methods, namely attention transfer on attention weights using soft-label cross-entropy and a square projections matrix $W_\phi \in \mathbb{R}^{h_d \times h_d}$. Our findings are illustrated in Figure 1, while per-task results are provided in Appendix A.5.2.

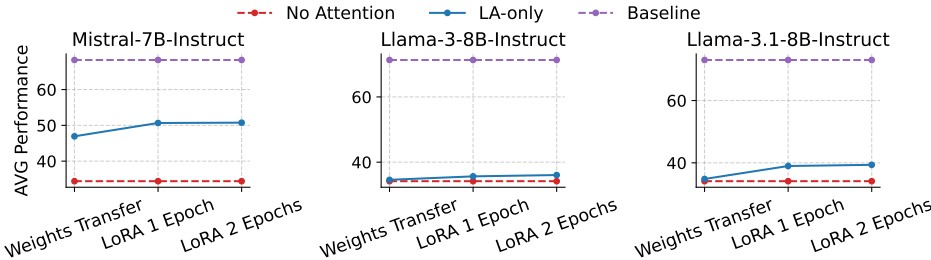

Figure 1: Measuring the performance of HedgeHog conversion at various points during the conversion process. LoRA refers to a LA-only fine-tuning.

We observe a clear contrast with the results in section 3.2, with the linear component in all models beating the no-attention ablation with no or little fine-tuning. However, we observe that the Llama models, and Llama-3 especially, only see a minor improvement in performance over no use of attention at all.

### 3.4 Component-Wise Investigation

Our minimal implementation of the HedgeHog conversion method successfully makes use of LA, although with mixed performance recovery. On the other hand, LoLCATs claims to recover base model performance, but appears to do so entirely using SWA. In this section, we seek to identify which parts of these two methods contribute to these two different results. To this end, we ablate

various parts of these methods within the attention transfer stage and evaluate them with LA-only. Focusing on this stage of the conversion process allows us to determine exactly which components work best with LA without affecting any weights shared with SWA during fine-tuning. We focus our ablations on the Mistral model as it saw the greatest difference in LA performance when changing the attention transfer and projections $\phi$ from LoLCATs' to HedgeHog's.

### 3.4.1 ATTENTION TRANSFER LEARNING

In this ablation, we keep the square HedgeHog feature map used in section 3.3 and simply ablate the Attention Transfer objective. We evaluate three modes: (1) The HedgeHog attention weights transfer using soft-label cross entropy loss between softmax attention weights and the Hedgehog attention weights (both quadratic cost) (2) The LoLCATs hybrid attention outputs transfer using MSE loss between full-context softmax attention outputs (quadratic cost) and the hybrid SWA + LA outputs (linear cost), and (3) the MSE between full softmax attention outputs (quadratic cost) and the full LA outputs (linear cost). We omit any LoRA fine-tuning in these evaluations to ensure we measure the direct impact of attention transfer only. Our findings are outlined in Table 2, while our results for all models are included in Appendix A.5.3.

| Transfer Objective | PIQA | ARC-E | ARC-C | HellaSwag | WG | MMLU | AVG | Rec. Perf |
|---|---|---|---|---|---|---|---|---|
| *Mistral-7B-Instruct* | 79.27 | 80.01 | 52.22 | 74.60 | 69.93 | 53.51 | 68.26 | 100.00 |
| Attention Weights | 67.90 | 61.74 | 29.69 | 45.82 | **52.64** | 23.82 | 46.94 | 68.76 |
| Attention Outputs | **69.64** | **62.46** | **31.40** | **46.61** | 50.28 | 23.00 | **47.23** | **69.20** |
| Hybrid Attention Out. | 55.44 | 32.83 | 24.06 | 27.36 | 49.80 | 22.98 | 35.41 | 51.88 |
| No Attention | 53.92 | 25.63 | 24.66 | 25.99 | 50.67 | **25.51** | 34.40 | 50.39 |

Table 2: Comparing the performance of LA-only after the attention transfer phase across learning objectives used during this stage.

Our results suggest that the hybrid attention output objective is likely to be responsible for LoLCATs failure to make use of LA, as it barely beats no attention at all. On the other hand, using a full attention output objective seems to beat even the weights transfer objective in this setting. In fact, this objective showed a particular advantage with Llama-3, although both Llama models still lag behind Mistral's recovered performance. We note that we have not observed any successful LA-only conversions of these models in the literature, as opposed to Mistral, suggesting that they may be particularly hard to convert. We leave the analysis as to why this might be to future work.

### 3.4.2 Φ DIMENSIONALITY

We now seek to measure the impact of the feature map size on conversion success. The original HedgeHog paper makes use of a square linear map $W_\phi \in \mathbb{R}^{h_d \times h_d}$ ($\phi : \mathbb{R}^{h_d} \to \mathbb{R}^{2h_d}$, given Eq. 4), with $h_d$: query-key head dimension, but LoLCATs reduce this to a rectangular linear map $W_\phi \in \mathbb{R}^{h_d \times \frac{h_d}{2}}$, for a significantly smaller $\phi : \mathbb{R}^{h_d} \to \mathbb{R}^{h_d}$. To this end, we repeat the experiments presented above in Table 2, with $W_\phi \in \mathbb{R}^{h_d \times \frac{h_d}{2}}$. Our findings are outlined in Table 3.

| Transfer Objective | PIQA | ARC-E | ARC-C | HellaSwag | WG | MMLU | AVG | Rec. Perf |
|---|---|---|---|---|---|---|---|---|
| *Mistral-7B-Instruct* | 79.27 | 80.01 | 52.22 | 74.60 | 69.93 | 53.51 | 68.26 | 100.00 |
| Attention Weights | **64.85** | **54.84** | 26.11 | **39.48** | **51.07** | 23.17 | **43.25** | **63.37** |
| Attention Outputs | 63.71 | 53.83 | **27.30** | 38.50 | 49.33 | 22.90 | 42.60 | 62.40 |
| Hybrid Attention Out. | 53.97 | 30.81 | 22.44 | 27.42 | 49.88 | 22.99 | 34.59 | 50.67 |
| No Attention | 53.92 | 25.63 | 24.66 | 25.99 | 50.67 | **25.51** | 34.40 | 50.39 |

Table 3: Repeating ablations described in Table 2, but with half the $W_\phi$ size.

Our results broadly follow that of section 3.4.1, with performance decreasing across the board with this reduction of output features. More specifically, we see the hybrid attention objective still improves performance only marginally, while the attention weights and outputs objectives achieve a significant and similar improvement on no attention. We note that the weights transfer objective appears to have the upper hand with this smaller $W_\phi$.

### 3.4.3 Φ ACTIVATION FUNCTION

Finally, HedgeHog implements an exponential activation function ($\sigma = \exp(\cdot)$). On the other hand, LoLCATs use the softmax, which is the layer-normalised equivalent ($\sigma = \text{softmax}(\cdot)$). Here, we ablate the two, and compare the resulting performance along with T2R and SUPRA's ReLU activation, as well as no activation function at all. Our findings are outlined in Table 4.

| Φ Activation Fn | PIQA | ARC-E | ARC-C | HellaSwag | WG | MMLU | AVG | Rec. Perf |
|---|---|---|---|---|---|---|---|---|
| *Mistral-7B-Instruct* | 79.27 | 80.01 | 52.22 | 74.60 | 69.93 | 53.51 | 68.26 | 100.00 |
| Softmax | **67.90** | **61.74** | **29.69** | **45.82** | **52.64** | 23.82 | **46.94** | **68.76** |
| Exponential | 66.32 | 61.45 | **29.69** | 44.67 | 50.99 | 23.52 | 46.11 | 67.55 |
| ReLU | 56.53 | 34.81 | 22.95 | 27.95 | 50.91 | 23.29 | 36.07 | 52.85 |
| 1+ ELU | 54.68 | 29.12 | 23.81 | 25.98 | 51.46 | **25.64** | 35.12 | 51.45 |
| None | 51.74 | 25.51 | 29.61 | 25.97 | 51.07 | 24.95 | 34.81 | 51.00 |
| No Attention | 53.92 | 25.63 | 24.66 | 25.99 | 50.67 | 25.51 | 34.40 | 50.39 |

Table 4: Ablating the activation function used in HedgeHog projections ($\phi$). All results show checkpoints for a single epoch of weights transfer with no fine-tuning, evaluated as LA-only.

Our results show the the exponential-based activation functions (exponential and softmax) far outperform the alternatives. In this regard, LoLCATs and HedgeHog appear to achieve comparable performance, with a slight edge in LoLCATs. Interestingly, 1+ELU does not appear to carry this same performance despite its close similarity to the exponential function by itself.

## 4 METHODS & RESULTS

Section 3 demonstrates some key issues in hybrid conversion methods, and isolates the components responsible for such shortcomings, namely the hybrid attention transfer objective and the dimensionality of $\phi$. In this section, we build on top of these findings and propose, evaluate, and compare three different methods to fine-tune a model for hybrid attention while avoiding the dominance of SWA and decay of LA. Note that all experiments conducted in this section use the same experimental setup as in Subsection 3.1, unless explicitly stated otherwise.

### 4.1 INFERENCE-TIME HYBRID

Seeing as fine-tuning for hybrid attention appears to encourage the model to focus on the more expressive SWA, we first propose a "zero-shot" hybrid, introducing SWA in models which have either seen no fine-tuning or fine-tuning with LA-only. We implement two modes of hybrid attention: one where LA only sees tokens outside of SWA's sliding window, and one where LA sees all past tokens therefore overlapping with SWA's context ("Overlap" in our results). Our findings are illustrated in Figure 2, while per-task results are provided in Appendix A.5.4.

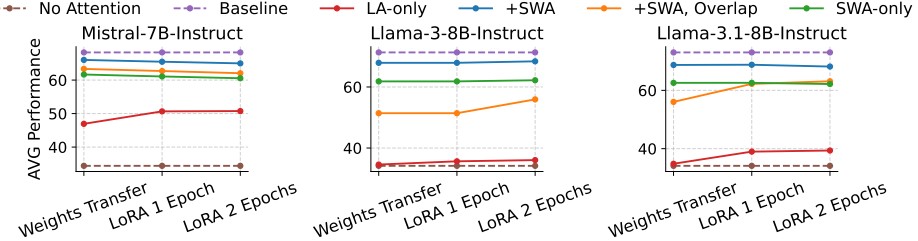

Figure 2: Adding SWA at inference time at various points of the conversion process.

We find that, while LA-only benefits from fine-tuning, the latter appears to slightly degrade the performance of SWA. As such, the best inference-time hybrid performance is achieved with no fine-tuning. Furthermore, it would appear as though overlapping LA's context with that of SWA also degrades performance. We theorise that this is most likely due to our fixed mixing term $g = 0.5$ and

may be improved by a data-dependent mixing term which may dynamically choose how to weight the contributions of each component in the case of overlap, as employed by Irie et al. (2025).

## 4.2 HEDGECATS: HEDGEHOG TRANSFER + LOLCATS FINE-TUNING

Our second method, HedgeCATs, blends HedgeHog's attention weight transfer with LoLCATs' LoRA fine-tuning of hybrid attention. The first training stage per-forms HedgeHog-style transfer to learn a feature map $\phi$ that mimics full soft-max attention, training with LA-only so the LA path stands up on its own.

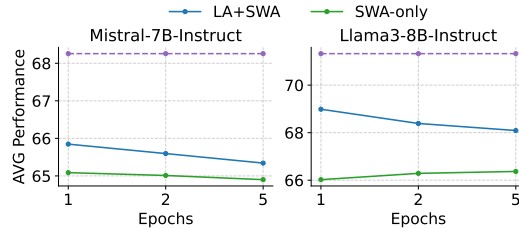

Figure 3: The average performance of HedgeCATs-trained models for difference amounts of LoRA fine-tuning.

Stage 2 applies LoRA fine-tuning while re-introducing SWA, aiming to recover base-model accuracy without letting the hybrid collapse back onto SWA. In practice, early stopping is key: Figure 3 shows that for Llama-3 8B, as fine-tuning proceeds the hybrid branch trends down, whereas SWA-only improves; for Mistral, both hybrid and SWA-only degrade with more LoRA steps. These behaviours suggest a very short LoRA schedule gives the best trade-off between accuracy, attributional validity of hybrid attention, and training budget. Per-task performance can be seen in Appendix A.5.5.

## 4.3 SSD: SCHEDULED SLIDING-WINDOW DROPOUT

Finally, to further guard against SWA dominance during training, we introduce Scheduled Sliding-window Dropout (SSD). SSD alters the SWA component in hybrid attention during LoRA fine-tuning according to a dropout and sliding-window size schedule, such that the model initially has to mostly make use of LA and gradually gains access to more SWA context and outputs across fine-tuning epochs. Our results across SSD fine-tuning regimes are summarised in Figure 4. We vary dropout and sliding-window schedules (either fixed or epoch-varying) and evaluate at multiple fine-tuning epochs to characterise performance as a function of training time. Scheduled parameters are applied per epoch: at epoch $k$ we use the $k$-th value in the schedule, and once the schedule is exhausted the final value is held fixed. This analysis illustrates how the SWA component of the hybrid attention mechanism evolves during fine-tuning. Figure 4a uses a dropout schedule ($0.9 \rightarrow 0.75 \rightarrow 0.5$), i.e., the SWA branch is dropped 90% of the time in epoch 1, with a fixed sliding-window size of 32; Figure 4b fixes dropout at 0.5 and schedules the sliding-window size ($4 \rightarrow 8 \rightarrow 16 \rightarrow 32 \rightarrow 64$); Figure 4c fixes both dropout and sliding-window size at 0.5 and 16, respectively. Per-task results for each experimented SSD setting are provided in Appendix A.5.6.

The results for SSD-trained models show consistent trends across Mistral and Llama-3. For Figure 4a, LA+SWA improves steadily with fine-tuning. This indicates that heavy early SWA dropout success-fully pushes learning through the linear path before relaxing to 0.5. In Figure 4b, performance starts lower compared to the previous experiment (penalised by the short initial window size) and remains flat or slightly down for Mistral, with only a mild late recovery for Llama-3; SWA-only also drifts down or flattens, suggesting limited benefit from widening the window alone in this setting. The fixed dropout and sliding-window model, Figure 4c, yields similar results to Figure 4a for Llama and slightly worse performance for Mistral. Across experiment settings, SWA-only sits well below Linear+SWA, confirming that our fine-tuning schedule leads to component-balanced hybrid attention.

## 5 DISCUSSION & CONCLUSION

Our analysis reveals a critical failure mode in hybrid attention conversion methods: models trained with hybrid objectives often bypass LA entirely, relying exclusively on SWA. In contrast, attention weights or full attention outputs transfer objectives successfully enable LA, though with varying performance recovery. We also confirm previous findings within our corrected framework, namely that exponential-family activations outperform alternatives, and larger feature map projections improve performance. These findings likely extend to other hybrid methods, like Liger, which report similar

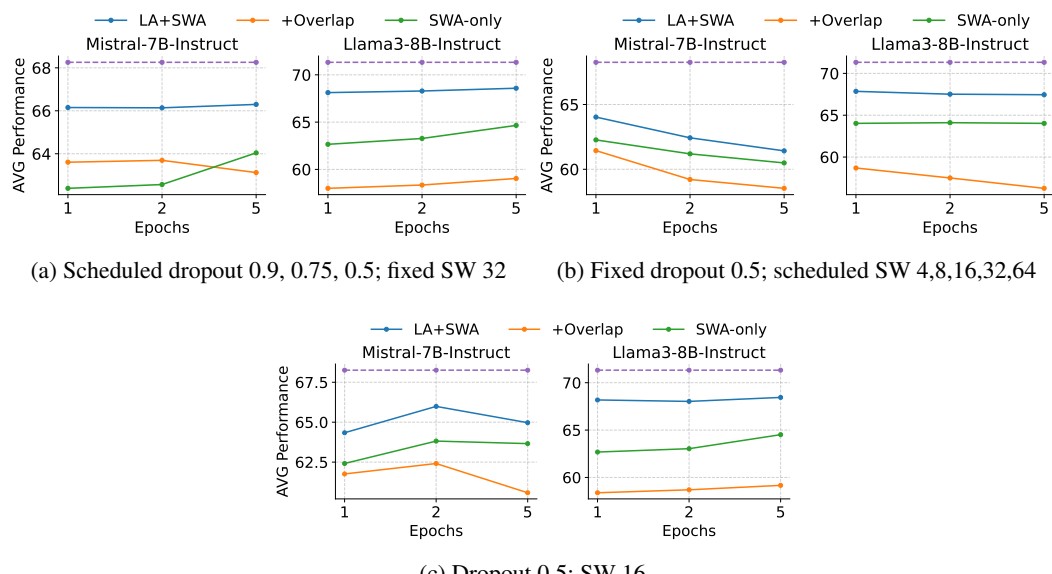

(a) Scheduled dropout 0.9, 0.75, 0.5; fixed SW 32     (b) Fixed dropout 0.5; scheduled SW 4,8,16,32,64

(c) Dropout 0.5; SW 16

Figure 4: Performance comparison of dropout and sliding-window size schedules for a different number of fine-tuning epochs. All resulting models are evaluated with a sliding-window size of $64$.

component imbalances. To address this issue, we proposed three solutions for converting a model to use hybrid attention: (1) Inference-time hybrid addition, which preserves LA without additional training; (2) HedgeCATs, combining successful weights transfer with careful early-stopping during hybrid fine-tuning; and (3) Scheduled Sliding-window Dropout (SSD), providing robust training that maintains component balance. Each offers different trade-offs between simplicity, individual component performance, and training cost, while all recovering over $95\%$ of base model performance. Importantly, the SWA-driven LoLCATs-trained hybrids (Table 1) consistently underperform compared to all our proposed methods. This indicates that enforcing genuine LA utilisation improves overall accuracy as well as attributional validity.

Finally, in an effort to offer clear guidance for future work to diagnose such issues in hybrids, we formalise our key findings surrounding component collapse into two complementary, inference-time ablation-based metrics. The first is essentially measured as the gap in downstream task-performance between the full hybrid model and SWA-only, wherein any significant increase in hybrid performance over SWA-only can be attributed to the LA component, regardless of LA-only performance. The second measures the same gap between the LA-only component and the complete removal of attention (No Attention (NA)), in order to quantify to what extent the linear component by itself is able to approximate softmax attention and provide outputs which are expected and useful within the rest of the model. We define them as follows:

$$\Delta_{H-SWA} := \text{Perf}_{Hybrid} - \text{Perf}_{SWA} \; ; \; \Delta_{LA-NA} := \text{Perf}_{LA} - \text{Perf}_{NA} \qquad (5)$$

Based on our results and intuition above, we would recommend a minimum of $\Delta_{H-SWA} \gg 0$, but ideally one would also want to observe $\Delta_{LA-NA} \gg 0$ to ensure meaningful LA outputs.

**Limitations and Future Work**    In this study, performance is limited by our simplified implementation of some of the components. For example, it should be noted that replacing our fixed mixing term $g$ with learned and dynamic mixing mechanisms, as seen in other methods (see section 2) is likely to increase performance. Although we motivate such choices with this study's focus on clear performance attribution between LA and SWA, while minimising the model's ability to discard LA, future work should extend our resulting methods to maximise performance through further investigation of components such as the mixing term, normalisation methods, training datasets, LoRA settings, etc. Finally, while further tuning the schedules used in SSD may lead to better performance,

it should be noted that such tuning is costly and therefore presents itself as a weakness which our other two proposed methods do not have.

LA is also believed to suffer from the same capacity issues observed in associative memory networks (Schlag et al., 2021). To this end, multiple gating mechanisms have been added to LA in order to manage information retention and retrieval accuracy across longer sequences (Schlag et al., 2021; Sun et al., 2023; Yang et al., 2023; 2024b; Liu et al., 2024a). Future work should examine how conversion methods may be applied to convert models to use such mechanisms. Additionally, LA conversion methods have the potential to improve performance of grouped KV retrieval methods (Xiao et al., 2024; Fountas et al., 2025), which reduce information dilution in long-context settings by only attending to subsets of past tokens. Such methods are limited by the memory requirements for storing long token histories. An interesting extension of this work would use LA to compress individual memory blocks, reducing memory requirements while limiting the capacity issues arising from LA being applied to entire long-context sequences.

**Conclusion**    In conclusion, while hybrid conversions promise efficiency with minimal performance loss, without careful design they fail to genuinely adopt LA. By identifying this failure mode and proposing solutions that maintain component balance, we restore attributional validity, ensuring claimed architectural components actually contribute to model behaviour, which is essential for advancing efficient Transformer architectures.

ETHICS STATEMENT

This work focuses on foundational methods for converting large language models to use LA mechanisms. All experiments were conducted using publicly-available pre-trained models, and widely used training and benchmark datasets. No personally identifiable or sensitive data was used. The primary contribution is methodological, and thus does not introduce new societal risks beyond those already known for these models, such as potential biases or misuse. We encourage responsible and transparent use of these methods.

REPRODUCIBILITY STATEMENT

We include experimental details across sections 2- 4, as well as Appendix A.2. Any details which aren't explicitly mentioned in this work are aligned with related works and clearly mentioned as such. Our experiments are reproducible within the LoLCATs public codebase with relatively minimal changes, and we are working on releasing our own public version of our codebase including all mentioned methods and ablations.

USE OF LARGE LANGUAGE MODELS

Our use of LLMs in the writing of this paper is limited to sparse and light improvements in wording within the main text.

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

# A APPENDIX

## A.1 NOTATION

**Vector operations**   Take vectors $x$, $y$

- $x^\top y$: vector inner-product
- $xy^\top$: vector outer-product

**Operators**

- $\odot$: Hadamard product
- $\oplus$: Concatenation

## A.2 IMPLEMENTATION DETAILS

Most of the experimental details are included across sections 2- 4. Any details which aren't explicitly mentioned either in theses sections or this one are aligned with related works and clearly mentioned as such.

### A.2.1 WEIGHTS INITIALISATION

$\Phi$   Linear projections, as in $\phi(\cdot)$, were initialised as the identity, following Zhang et al. (2024b), with added Gaussian-sampled noise ($\mu = 0$ and $\sigma = 0.1$) using a seed-based generator to ensure the same initialisation across all runs.

## A.3 EXTENDED RELATED WORKS

| Method | Attn Type | Feature map $\phi(x)$ | Transfer objective |
|--------|-----------|------------------------|--------------------|
| **T2R/SUPRA** | Fully linear | $\mathrm{ReLU}(W_\phi^\top x + b)$ | Uptrain with standard causal LM cross-entropy (no attention-distillation objective). |
| **HedgeHog** | Fully linear | $\exp(W_\phi^\top x + b)$* | Minimises cross-entropy / KL between softmax attention weights and linear weights. |
| **LoLCATs** | Hybrid (LA + SWA) | $\mathrm{Softmax}(W_\phi^\top x + b)$* | Minimises MSE between softmax attention outputs and linear (hybrid) attention outputs. |
| **Liger** | Hybrid (GLA + SWA) | $\mathrm{Softmax}(x)$ | Causal LM cross-entropy with LoRA fine-tuning (no attention-distillation objective). |

Table 5: Side-by-side comparison of linearising conversions for pre-trained Transformers. *Methods for which the negative mapping is concatenated to the final output of $\phi$ (see Eq. 4).

### A.3.1 LONG-CONTEXT PERFORMANCE

Linear attention mechanisms are often motivated by their computational advantages in long-context scenarios due to $\mathcal{O}(n)$ complexity. However, there is mounting evidence that vanilla linear attention - the form we study in this paper - is fragile in genuinely long-context regimes. Many linear attention papers either avoid long-context evaluation altogether (Katharopoulos et al., 2020; Kasai et al., 2021; Wang et al., 2020) or report only perplexity-based metrics (Sun et al., 2023; De et al., 2024; Gu & Dao, 2023). More complex variants, such as Gated DeltaNet (Yang et al., 2024b) and other gated variants (Yang et al., 2023; Sun et al., 2023; Liu et al., 2024a; Behrouz et al., 2024), motivate their use of state-decay factors, gated updates, and learned mixing terms as ways to manage the shortcomings of linear attention in these settings, making long-context evaluation more appropriate for

such architectures. However, for vanilla linear attention, strong long-context performance typically requires either explicit training at the target sequence length or - as is the case with softmax attention - dedicated context-extrapolation mechanisms (e.g., RoPE scaling (Liu et al., 2024b) or YaRN-style methods (Peng et al., 2024)).

This fragility extends to post-hoc conversion methods that introduce linear attention into pre-trained models. LoLCATs and related conversion approaches (Lan et al., 2025; Kasai et al., 2021) are empirically validated primarily in short-context regimes. When these methods report long-context experiments (e.g., passkey retrieval), they require additional training at the target sequence length, effectively treating long-context performance as a separate setting rather than a direct consequence of the conversion itself. While SUPRA (Mercat et al., 2024) reports long-context results without explicit retraining at longer sequence lengths, the authors explicitly note that Mistral-SUPRA's apparent robustness at large context lengths stems largely from added decay factors that progressively down-weight distant tokens, thereby shortening the effective context length. In other words, SUPRA operates with a relatively short receptive field within a longer context window rather than demonstrating genuine long-context utilization.

Our work is explicitly focused on understanding and correcting the failure modes of LoLCATs-style hybrid conversions in their original short-context regime. We diagnose why the linear component collapses and propose interventions that restore balanced component usage under the same training and evaluation conditions as the original LoLCATs validation. For completeness, we conducted a small set of long-context experiments with our converted models (Appendix A.6.6). However, these results sit in the under-specified regime described above: they lack the additional long-context training or dedicated extrapolation machinery that current work deems necessary for reliable performance. Consequently, they provide little reliable signal about the relative quality of different conversion strategies and do not affect our conclusions about component imbalance. For these reasons, we restrict our main empirical analysis to the short-context setting in which LoLCATs and related conversion methods are originally validated, leaving a systematic study of how our interventions interact with long-context training and extrapolation techniques to future work.

### A.4    COMPLEXITY ANALYSIS

Hydrid linear attention, as noted in previous work (Zhang et al., 2024a; Beltagy et al., 2020; Arora et al., 2024; Munkhdalai et al., 2024), grows linearly with sequence length $n$. This offers a significant improvement over the quadratic $\mathcal{O}(n^2 d)$ time and $\mathcal{O}(n^2)$ space complexity of Softmax attention Vaswani et al. (2017). Hybrid linear attention has a complexity of $\mathcal{O}(n(d^2 + wd))$ where $d$ is the hidden dimension, and $w$ is the sliding-window size. Regarding our proposed methods, they all incur the same inference-time computational cost as LoLCATs models. During training, our methods have the same theoretical computational training cost as LoLCATs. For attention transfer, while weight transfer causes the linear component to incur quadratic complexity during conversion, this does not affect the asymptotic $\mathcal{O}$ time complexity as the target softmax attention output already incurs quadratic cost. However, in practice, it does result in slightly increased wall-clock time over the attention output transfer objective. As for the fine-tuning stage, LA-only fine-tuning used with inference-time hybrids reduces to $\mathcal{O}(nd^2)$, while SSD holds the same complexity as LoLCATs but may see lower wall-clock times in the case of smaller, scheduled sliding-windows. Finally, HedgeCATs does not change this stage.

We empirically measure efficiency metrics during evaluation in long-context to show how hybrid linear attention permits for long sequence lengths as opposed to getting an out-of-memory (OOM) error as with eager softmax attention (Table 6). Note that our linear attention component has not been optimised with a custom kernel, and that using kernel-optimised softmax attention such as FlashAttention-2 Dao (2023) will expectedly, and perhaps misleadingly, result in better efficiency.

| Model | Attn Module | Eval Wall-Time (mins) | GPU Peak Mem (GiB) | Samples per Second |
|---|---|---|---|---|
| Mistral-7B-Instruct | | | | |
| | Hybrid: SWA+LA | 310.19 | 79.27 | 0.30 |
| | Softmax (FlashAttn-2) | 163.3 | 33.40 | 0.58 |
| | Softmax (Eager) | OOM | OOM | OOM |
| Llama3-8B-Instruct | | | | |
| | Hybrid: SWA+LA | 554.23 | 78.02 | 0.17 |
| | Softmax (FlashAttn-2) | 133.76 | 31.68 | 0.70 |
| | Softmax (Eager) | OOM | OOM | OOM |

Table 6: Long-context (SCROLLS Shaham et al. (2022)) evaluation efficiency of hybrid linear attention (unoptimised) and softmax attention using both optimised (FlashAttention-2) and unoptimised (eager) kernels.

## A.5 FURTHER RESULTS

In this section, we provide more detailed results for the experiments presented in the main text.

### A.5.1 ATTENTION COMPONENTS ABLATION WITH PER-TASK STANDARD ERRORS

| Active Attn Modules | PIQA | ARC-E | ARC-C | HellaSwag | WG | MMLU | AVG | Rec. Perf |
|---|---|---|---|---|---|---|---|---|
| *Mistral-7B-Instruct* | 79.27 | 80.01 | 52.22 | 74.60 | 69.93 | 53.51 | 68.26 | 100.00 |
| SWA + Linear | **78.24** ± 0.19 | **79.59** ± 0.13 | **50.42** ± 0.27 | 71.19 ± 0.05 | 68.09 ± 0.19 | 45.71 ± 0.17 | 65.54 ± 0.07 | 96.02 ±0.1 |
| SWA only | 78.02 ± 0.29 | 79.46 ± 0.02 | 49.77 ± 0.06 | 68.28 ± 1.15 | **68.51** ± 0.24 | 46.90 ± 0.17 | 65.16 ± 0.21 | 95.46 ±0.3 |
| Linear only | 53.75 ± 0.06 | 29.73 ± 0.42 | 25.08 ± 0.32 | 27.21 ± 0.11 | 50.54 ± 0.37 | 23.12 ± 0.08 | 34.91 ± 0.07 | 51.14 ± 0.1 |
| SWA + Attn sinks | 78.17 ± 0.04 | 79.50 ± 0.11 | 50.17 ± 0.13 | **73.37** ± 0.02 | 68.38 ± 0.22 | **48.66** ± 0.13 | **66.38** ± 0.08 | **97.24** ± 0.12 |
| Attn sinks only | 54.41 | 30.05 | 24.66 | 28.80 | 49.64 | 23.67 | 35.21 | 51.58 |
| No Attention | 53.92 ± 0.00 | 25.63 ± 0.00 | 24.66 ± 0.00 | 25.99 ± 0.00 | 50.67 ± 0.00 | 25.51 ± 0.00 | 34.40 ± 0.00 | 50.39 ± 0.00 |
| *Llama-3-8B-Instruct* | 78.13 | 81.69 | 56.66 | 75.94 | 71.67 | 63.85 | 71.32 | 100.00 |
| SWA + Linear | 78.02 ± 0.16 | 80.15 ± 0.16 | 53.78 ± 0.30 | 71.93 ± 0.16 | 71.85 ± 0.22 | 48.78 ± 0.24 | 67.42 ± 0.07 | 94.53 ± 0.09 |
| SWA only | 78.09 ± 0.16 | 80.00 ± 0.01 | 54.01 ± 0.05 | 64.93 ± 0.09 | 71.88 ± 0.28 | 47.05 ± 0.16 | 65.99 ± 0.05 | 92.52 ± 0.06 |
| Linear only | 52.90 ± 0.44 | 25.71 ± 0.32 | 26.02 ± 0.30 | 26.47 ± 0.24 | 48.72 ± 0.30 | 22.96 ± 0.01 | 33.80 ± 0.15 | 47.39 ± 0.20 |
| SWA + Attn sinks | **78.33** ± 0.13 | **80.68** ± 0.06 | **54.52** ± 0.13 | **76.55** ± 0.05 | **71.95** ± 0.05 | **58.00** ± 0.29 | **70.01** ± 0.04 | **98.15** ± 0.06 |
| Attn sinks only | 55.40 *pm* 0.59 | 30.40 ± 0.72 | 22.61 ± 0.29 | 28.99 ± 0.13 | 50.78 ± 0.68 | 22.95 ± 0.01 | 35.19 ± 0.18 | 49.34 ± 0.25 |
| No Attention | 55.17 ± 0.00 | 26.73 ± 0.00 | 22.87 ± 0.00 | 26.13 ± 0.00 | 51.14 ± 0.00 | 22.95 ± 0.00 | 34.17 ± 0.00 | 47.91 ± 0.00 |

Table 7: Measuring the effect of attention components on benchmark accuracy for models trained with LoLCATs hybrid conversion. Standard errors ($N = 3$) are shown for the main ablations.

### A.5.2 LINEAR ATTENTION ONLY

| Active Attn Modules | PIQA | ARC-E | ARC-C | HellaSwag | WG | MMLU | AVG | Rec. Perf |
|---|---|---|---|---|---|---|---|---|
| *Mistral-7B-Instruct* | 79.27 | 80.01 | 52.22 | 74.60 | 69.93 | 53.51 | 68.26 | 100.00 |
| No Attention | 53.92 | 25.63 | 24.66 | 25.99 | 50.67 | **25.51** | 34.40 | 50.39 |
| LA Weights Transfer | 67.90 | 61.74 | 29.69 | 45.82 | **52.64** | 23.82 | 46.94 | 68.76 |
| +LoRA 1 Epoch | 70.13 | 66.20 | **34.90** | 56.39 | 51.38 | 24.95 | 50.66 | 74.22 |
| +LoRA 2 Epochs | **70.29** | **66.29** | 34.39 | **56.41** | 52.17 | 24.97 | **50.75** | **74.36** |
| *Llama3-8B-Instruct* | 78.13 | 81.69 | 56.66 | 75.94 | 71.67 | 63.85 | 71.32 | 100.00 |
| No Attention | **55.17** | 26.73 | 22.87 | 26.13 | **51.14** | 22.95 | 34.17 | 47.90 |
| LA Weights Transfer | 54.24 | 26.94 | **26.11** | 26.42 | 50.43 | **23.42** | 34.59 | 48.50 |
| +LoRA 1 Epoch | 54.90 | 29.46 | 25.68 | 32.06 | 49.01 | 22.79 | 35.65 | 49.98 |
| +LoRA 2 Epochs | 54.73 | **30.26** | 25.43 | **33.28** | 49.64 | 22.94 | **36.05** | **50.54** |
| *Llama3.1-8B-Instruct* | 80.14 | 81.82 | 55.20 | 79.14 | 73.72 | 68.05 | 73.01 | 100.00 |
| No Attention | 53.59 | 26.89 | **25.94** | 26.19 | 49.09 | 22.95 | 34.11 | 46.72 |
| LA Weights Transfer | 54.41 | 28.16 | 25.34 | 27.50 | 48.46 | **25.04** | 34.82 | 47.69 |
| +LoRA 1 Epoch | 58.71 | 38.51 | 24.23 | 35.38 | **53.51** | 23.53 | 38.98 | 53.39 |
| +LoRA 2 Epochs | **58.87** | **39.69** | 23.89 | **37.41** | 52.49 | 23.78 | **39.36** | **53.90** |

Table 8: Measuring the performance of HedgeHog conversion at various points during the conversion process. LoRA refers to a LA-only finetuning.

### A.5.3 ATTENTION TRANSFER LEARNING OBJECTIVE

| Transfer Objective | PIQA | ARC-E | ARC-C | HellaSwag | WG | MMLU | AVG | Rec. Perf |
|---|---|---|---|---|---|---|---|---|
| *Mistral-7B-Instruct* | 79.27 | 80.01 | 52.22 | 74.60 | 69.93 | 53.51 | 68.26 | 100.00 |
| Attention Weights | 67.90 | 61.74 | 29.69 | 45.82 | **52.64** | 23.82 | 46.94 | 68.76 |
| Attention Outputs | **69.64** | **62.46** | **31.40** | **46.61** | 50.28 | 23.00 | **47.23** | **69.20** |
| Hybrid Attention Out. | 55.44 | 32.83 | 24.06 | 27.36 | 49.80 | 22.98 | 35.41 | 51.88 |
| No Attention | 53.92 | 25.63 | 24.66 | 25.99 | 50.67 | **25.51** | 34.40 | 50.39 |
| *Llama3-8B-Instruct* | 78.13 | 81.69 | 56.66 | 75.94 | 71.67 | 63.85 | 71.32 | 100.00 |
| Attention Weights | 54.24 | 26.94 | **26.11** | 26.42 | 50.43 | 23.42 | 34.59 | 48.50 |
| Attention Outputs | **61.04** | **41.04** | 24.06 | **30.02** | 49.25 | **23.88** | **38.22** | **53.58** |
| Hybrid Attention Out. | 53.10 | 26.22 | 24.74 | 26.23 | 49.41 | 23.43 | 33.86 | 47.47 |
| No Attention | 55.17 | 26.73 | 22.87 | 26.13 | **51.14** | 22.95 | 34.17 | 47.90 |
| *Llama3.1-8B-Instruct* | 80.14 | 81.82 | 55.20 | 79.14 | 73.72 | 68.05 | 73.01 | 100.00 |
| Attention Weights | 54.41 | 28.16 | 25.34 | 27.50 | 48.46 | 25.04 | 34.82 | 47.69 |
| Attention Outputs | **54.90** | **28.49** | 25.60 | **28.60** | 48.78 | **26.30** | **35.45** | **48.55** |
| Hybrid Attention Out. | 54.57 | 25.46 | **26.02** | 26.04 | 47.75 | 23.37 | 33.87 | 46.39 |
| No Attention | 53.59 | 26.89 | 25.94 | 26.19 | **49.09** | 22.95 | 34.11 | 46.72 |

Table 9: Comparing the performance of linear attention only after the attention transfer phase across learning objectives used during this stage.

### A.5.4    ADDING SWA AT INFERENCE TIME

| Active Attn Modules | PIQA | ARC-E | ARC-C | HellaSwag | WG | MMLU | AVG | Rec. Perf |
|---|---|---|---|---|---|---|---|---|
| *Mistral-7B-Instruct* | 79.27 | 80.01 | 52.22 | 74.60 | 69.93 | 53.51 | 68.26 | 100.00 |
| LA Weights Transfer | 67.90 | 61.74 | 29.69 | 45.82 | 52.64 | 23.82 | 46.94 | 68.76 |
| *+SWA* | **79.11** | 79.76 | 51.62 | **70.57** | 69.93 | **45.13** | **66.02** | **96.72** |
| *+SWA, Overlap* | 77.48 | 78.79 | 50.77 | 67.13 | 61.96 | 43.88 | 63.34 | 92.79 |
| *SWA-only* | 77.97 | **79.92** | 51.02 | 51.19 | 69.93 | 39.92 | 61.66 | 90.33 |
| +LoRA 1 Epoch | 70.13 | 66.20 | 34.90 | 56.39 | 51.38 | 24.95 | 50.66 | 74.22 |
| *+SWA* | 78.73 | 78.83 | **51.96** | 70.44 | **70.24** | 42.66 | 65.48 | 95.93 |
| *+SWA, Overlap* | 77.42 | 78.20 | 49.06 | 67.38 | 63.30 | 40.84 | 62.70 | 91.86 |
| *SWA-only* | 77.91 | 78.75 | 51.54 | 48.63 | **70.24** | 39.41 | 61.08 | 89.49 |
| +LoRA 2 Epochs | 70.29 | 66.29 | 34.39 | 56.41 | 52.17 | 24.97 | 50.75 | 74.36 |
| *+SWA* | 78.62 | 78.70 | 51.11 | 69.75 | 69.61 | 42.14 | 64.99 | 95.21 |
| *+SWA, Overlap* | 77.26 | 77.78 | 48.04 | 66.22 | 62.98 | 39.99 | 62.05 | 90.90 |
| *SWA-only* | 77.80 | 78.49 | 50.77 | 47.69 | 69.61 | 38.91 | 60.55 | 88.70 |
| *Llama3-8B-Instruct* | 78.13 | 81.69 | 56.66 | 75.94 | 71.67 | 63.85 | 71.32 | 100.00 |
| LA Weights Transfer | 54.24 | 26.94 | 26.11 | 26.42 | 50.43 | 23.42 | 34.59 | 48.50 |
| *+SWA* | 77.80 | **81.06** | **55.97** | 71.71 | 71.67 | 49.22 | 67.91 | 95.21 |
| *+SWA, Overlap* | 67.03 | 56.06 | 36.86 | 49.67 | 58.56 | 40.26 | 51.41 | 72.08 |
| *SWA-only* | 76.28 | 80.09 | 55.12 | 44.48 | 71.67 | 43.31 | 61.83 | 86.68 |
| +LoRA 1 Epoch | 54.90 | 29.46 | 25.68 | 32.06 | 49.01 | 22.79 | 35.65 | 49.98 |
| *+SWA* | 77.80 | **81.06** | **55.97** | 71.71 | 71.67 | 49.22 | 67.91 | 95.90 |
| *+SWA, Overlap* | 67.03 | 56.06 | 36.86 | 49.67 | 58.56 | 40.26 | 51.41 | 78.42 |
| *SWA-only* | 76.28 | 80.09 | 55.12 | 44.48 | 71.67 | 43.31 | 61.83 | 87.20 |
| +LoRA 2 Epochs | 54.73 | 30.26 | 25.43 | 33.28 | 49.64 | 22.94 | 36.05 | 50.54 |
| *+SWA* | **78.35** | 80.22 | 55.55 | **73.15** | **73.24** | **49.87** | **68.40** | **95.27** |
| *+SWA, Overlap* | 70.62 | 61.20 | 41.13 | 58.61 | 60.30 | 43.71 | 55.93 | 78.10 |
| *SWA-only* | 76.99 | 79.46 | 54.01 | 45.71 | **73.24** | 43.77 | 62.20 | 86.74 |
| *Llama3.1-8B-Instruct* | 80.14 | 81.82 | 55.20 | 79.14 | 73.72 | 68.05 | 73.01 | 100.00 |
| LA Weights Transfer | 54.41 | 28.16 | 25.34 | 27.50 | 48.46 | 25.04 | 34.82 | 47.69 |
| *+SWA* | 79.87 | 80.98 | **54.78** | **72.81** | 73.72 | 50.11 | 68.71 | 94.11 |
| *+SWA, Overlap* | 71.38 | 64.35 | 40.44 | 51.66 | 62.51 | 45.90 | 56.04 | 76.75 |
| *SWA-only* | 78.51 | 80.35 | 53.24 | 46.08 | 73.72 | 43.51 | 62.57 | 85.70 |
| +LoRA 1 Epoch | 58.71 | 38.51 | 24.23 | 35.38 | 53.51 | 23.53 | 38.98 | 53.39 |
| *+SWA* | **80.09** | **81.44** | 53.58 | 72.77 | **74.03** | **50.76** | **68.78** | **94.20** |
| *+SWA, Overlap* | 76.88 | 75.93 | 48.89 | 61.16 | 62.59 | 47.79 | 62.21 | 85.20 |
| *SWA-only* | 78.94 | 80.64 | 51.96 | 46.24 | **74.03** | 43.64 | 62.58 | 85.71 |
| +LoRA 2 Epochs | 58.87 | 39.69 | 23.89 | 37.41 | 52.49 | 23.78 | 39.36 | 53.90 |
| *+SWA* | 79.65 | 80.72 | 53.58 | 72.07 | 73.40 | 49.56 | 68.16 | 93.36 |
| *+SWA, Overlap* | 77.48 | 77.82 | 50.51 | 61.46 | 63.38 | 47.98 | 63.11 | 86.43 |
| *SWA-only* | 78.45 | 79.92 | 52.39 | 45.27 | 73.40 | 43.58 | 62.17 | 85.15 |

Table 10: Adding SWA at inference time at various points of the conversion process.

### A.5.5 HEDGECATS

| Active Attn Module | PIQA | ARC-E | ARC-C | HellaSwag | WG | MMLU | AVG | Rec. Perf |
|---|---|---|---|---|---|---|---|---|
| *Mistral-7B-Instruct* | 79.27 | 80.01 | 52.22 | 74.60 | 69.93 | 53.51 | 68.26 | 100.00 |
| 1 epoch | | | | | | | | |
| Linear+SWA | 78.51 | 79.46 | **51.45** | **71.14** | **69.38** | 45.14 | **65.85** | **96.47** |
| SWA-only | 78.24 | 79.34 | 50.68 | 65.92 | **69.38** | 46.96 | 65.09 | 95.36 |
| 2 epochs | | | | | | | | |
| Linear+SWA | 78.24 | 79.59 | 51.02 | 70.73 | 68.75 | 45.24 | 65.60 | 96.10 |
| SWA-only | 78.07 | 79.71 | 50.17 | 66.30 | 68.75 | 47.07 | 65.01 | 95.25 |
| 5 epochs | | | | | | | | |
| Linear+SWA | **78.56** | 79.42 | 50.26 | 70.55 | 67.64 | 45.62 | 65.34 | 95.73 |
| SWA-only | 78.51 | **79.67** | 49.66 | 66.51 | 67.64 | **47.41** | 64.90 | 95.08 |
| *Llama3-8B-Instruct* | 78.13 | 81.69 | 56.66 | 75.94 | 71.67 | 63.85 | 71.32 | 100.00 |
| 1 epoch | | | | | | | | |
| Linear+SWA | **79.38** | **81.78** | **55.97** | **72.28** | 73.32 | 51.18 | **68.99** | **96.72** |
| SWA-only | 78.89 | 81.61 | 55.63 | 59.67 | 73.32 | 47.02 | 66.02 | 92.57 |
| 2 epochs | | | | | | | | |
| Linear+SWA | 78.62 | 80.89 | 53.92 | 71.54 | **73.48** | 51.87 | 68.39 | 95.88 |
| SWA-only | 78.40 | 81.14 | 54.27 | 62.68 | **73.48** | 47.76 | 66.29 | 92.94 |
| 5 epochs | | | | | | | | |
| Linear+SWA | 78.73 | 80.05 | 52.99 | 71.49 | 72.38 | **52.90** | 68.09 | 95.47 |
| SWA-only | 78.78 | 80.39 | 53.33 | 65.57 | 72.38 | 47.76 | 66.37 | 93.05 |

Table 11: HedgeCATs performance for different activated attention modules at inference time evaluated at different stages of LoRA finetuning.

## A.5.6 SSD: TESTING SCHEDULED HYBRID CONVERSION

| Active Attn Modules | PIQA | ARC-E | ARC-C | HellaSwag | WG | MMLU | AVG | Rec. Perf |
|---|---|---|---|---|---|---|---|---|
| *Mistral-7B-Instruct* | 79.27 | 80.01 | 52.22 | 74.60 | 69.93 | 53.51 | 68.26 | 100.00 |
| 1 epoch | | | | | | | | |
| Linear+SWA | **78.94** | 79.55 | 52.47 | 71.55 | 69.61 | 44.79 | 66.15 | 96.92 |
| + Overlap | 77.86 | 78.58 | 50.34 | 68.46 | 63.61 | 42.79 | 63.61 | 93.19 |
| SWA-only | 78.24 | 79.59 | 52.39 | 53.77 | 69.61 | 40.74 | 62.39 | 91.40 |
| 2 epochs | | | | | | | | |
| Linear+SWA | 78.62 | **79.59** | 52.90 | **71.61** | 68.98 | **45.13** | 66.14 | 96.90 |
| + Overlap | 77.64 | 78.49 | 50.43 | 68.55 | 64.25 | 42.80 | 63.69 | 93.31 |
| SWA-only | 77.80 | 79.76 | 52.39 | 55.45 | 68.98 | 41.02 | 62.57 | 91.66 |
| 5 epochs | | | | | | | | |
| Linear+SWA | 78.62 | 79.55 | **53.58** | 71.06 | **69.93** | 45.04 | **66.30** | **97.13** |
| + Overlap | 77.48 | 77.90 | 50.68 | 68.09 | 62.35 | 42.24 | 63.12 | 92.48 |
| SWA-only | 78.40 | 79.25 | 53.24 | 61.45 | 69.93 | 41.97 | 64.04 | 93.82 |
| *Llama3-8B-Instruct* | 78.13 | 81.69 | 56.66 | 75.94 | 71.67 | 63.85 | 71.32 | 100.00 |
| 1 epoch | | | | | | | | |
| Linear+SWA | **78.40** | 80.72 | 55.72 | 72.07 | 71.43 | 50.38 | 68.12 | 95.59 |
| + Overlap | 70.89 | 63.80 | 41.81 | 63.53 | 59.98 | 47.98 | 58.00 | 81.32 |
| SWA-only | 77.26 | 79.88 | 54.61 | 48.04 | 71.43 | 44.70 | 62.65 | 87.84 |
| 2 epochs | | | | | | | | |
| Linear+SWA | 78.18 | **80.85** | 55.80 | 72.34 | 72.22 | 50.34 | 68.29 | 95.74 |
| + Overlap | 71.33 | 63.64 | 41.47 | 65.52 | 59.98 | 48.15 | 58.35 | 82.29 |
| SWA-only | 77.15 | 80.05 | 54.61 | 50.31 | 72.22 | 45.28 | 63.27 | 88.38 |
| 5 epochs | | | | | | | | |
| Linear+SWA | 78.13 | 80.72 | **56.23** | **72.84** | **72.61** | 50.97 | **68.58** | **95.97** |
| + Overlap | 71.55 | 63.30 | 41.13 | 68.13 | 60.38 | 49.75 | 59.04 | 82.78 |
| SWA-only | 77.48 | 80.43 | 55.12 | 56.11 | 72.61 | 46.16 | 64.65 | 90.65 |

Table 12: Finetuning using SSD for a dropout schedule of 0.9, 0.75, 0.5 and a fixed sliding window size of 32.

| Active Attn Modules | PIQA | ARC-E | ARC-C | HellaSwag | WG | MMLU | AVG | Rec. Perf |
|---|---|---|---|---|---|---|---|---|
| *Mistral-7B-Instruct* | 79.27 | 80.01 | 52.22 | 74.60 | 69.93 | 53.51 | 68.26 | 100.00 |
| 1 epoch | | | | | | | | |
|   Linear+SWA | 78.24 | 77.95 | 50.34 | 67.30 | 68.27 | 43.93 | 64.34 | 94.26 |
|   + Overlap | 77.37 | 77.19 | 47.10 | 66.63 | 59.91 | 42.36 | 61.76 | 90.48 |
|   SWA-only | 77.69 | 78.28 | 50.00 | 58.59 | 68.27 | 41.65 | 62.41 | 91.44 |
| 2 epochs | | | | | | | | |
|   Linear+SWA | **78.78** | 79.34 | **53.33** | **70.47** | 69.06 | **44.95** | **65.99** | **96.68** |
|   + Overlap | 77.31 | 77.44 | 48.46 | 66.74 | 61.48 | 43.04 | 62.41 | 91.44 |
|   SWA-only | 78.29 | **79.46** | 52.56 | 61.13 | 69.06 | 42.40 | 63.82 | 93.50 |
| 5 epochs | | | | | | | | |
|   Linear+SWA | 78.51 | 79.12 | 52.13 | 68.30 | 69.06 | 42.69 | 64.97 | 95.18 |
|   + Overlap | 76.82 | 76.52 | 47.95 | 62.36 | 60.54 | 39.30 | 60.58 | 88.76 |
|   SWA-only | 78.18 | 79.08 | 51.62 | 62.49 | 69.06 | 41.52 | 63.66 | 93.26 |
| *Llama3-8B-Instruct* | 78.13 | 81.69 | 56.66 | 75.94 | 71.67 | 63.85 | 71.32 | 100.00 |
| 1 epoch | | | | | | | | |
|   Linear+SWA | 78.07 | 80.64 | 55.38 | 72.23 | **72.38** | 50.38 | 68.18 | 95.59 |
|   + Overlap | 71.16 | 65.03 | 42.15 | 63.66 | 60.22 | 48.08 | 58.38 | 81.86 |
|   SWA-only | 76.88 | 79.76 | 54.27 | 48.21 | 72.38 | 44.59 | 62.68 | 87.88 |
| 2 epochs | | | | | | | | |
|   Linear+SWA | 78.07 | **80.77** | 54.86 | 72.41 | 71.59 | 50.46 | 68.03 | 95.38 |
|   + Overlap | 71.38 | 64.56 | 41.55 | 65.71 | 60.77 | 48.20 | 58.70 | 82.29 |
|   SWA-only | 77.04 | 80.05 | 53.75 | 50.37 | 71.59 | 45.41 | 63.04 | 88.38 |
| 5 epochs | | | | | | | | |
|   Linear+SWA | **78.35** | 80.43 | **55.97** | **72.89** | 72.14 | **50.90** | **68.45** | **95.97** |
|   + Overlap | 71.38 | 63.47 | 42.06 | 68.12 | 60.30 | 49.65 | 59.16 | 82.95 |
|   SWA-only | 77.64 | 80.18 | 54.86 | 56.07 | 72.14 | 46.20 | 64.52 | 90.45 |

Table 13: Finetuning using SSD for a fixed dropout of 0.5 and sliding window size of 16.

| Active Attn Modules | PIQA | ARC-E | ARC-C | HellaSwag | WG | MMLU | AVG | Rec. Perf |
|---|---|---|---|---|---|---|---|---|
| *Mistral-7B-Instruct* | 79.27 | 80.01 | 52.22 | 74.60 | 69.93 | 53.51 | 68.26 | 100.00 |
| 1 epoch | | | | | | | | |
|   Linear+SWA | **77.75** | 77.74 | **50.17** | **66.81** | 68.43 | **43.28** | **64.03** | **93.81** |
|   + Overlap | 77.26 | 76.56 | 46.67 | 66.01 | 59.91 | 42.24 | 61.44 | 90.02 |
|   SWA-only | 77.42 | **77.99** | 49.74 | 58.31 | 68.43 | 41.71 | 62.27 | 91.22 |
| 2 epochs | | | | | | | | |
|   Linear+SWA | 76.82 | 74.07 | 46.08 | 66.44 | 68.43 | 42.73 | 62.43 | 91.46 |
|   + Overlap | 75.84 | 72.31 | 40.96 | 65.79 | 58.88 | 41.48 | 59.21 | 86.75 |
|   SWA-only | 76.82 | 74.71 | 46.50 | 58.98 | 68.43 | 41.70 | 61.19 | 89.65 |
| 5 epochs | | | | | | | | |
|   Linear+SWA | 76.33 | 73.65 | 46.08 | 63.96 | 67.17 | 41.34 | 61.42 | 89.99 |
|   + Overlap | 75.63 | 72.31 | 42.41 | 62.89 | 59.27 | 38.66 | 58.53 | 85.75 |
|   SWA-only | 76.22 | 73.91 | 46.25 | 58.38 | 67.17 | 41.01 | 60.49 | 88.62 |
| *Llama3-8B-Instruct* | 78.13 | 81.69 | 56.66 | 75.94 | 71.67 | 63.85 | 71.32 | 100.00 |
| 1 epoch | | | | | | | | |
|   Linear+SWA | 77.91 | **80.39** | **54.27** | 71.49 | 72.06 | **51.00** | **67.85** | **95.13** |
|   + Overlap | 71.00 | 63.64 | 40.87 | 66.37 | 61.40 | 48.88 | 58.69 | 82.29 |
|   SWA-only | 77.26 | 80.13 | 53.75 | 55.03 | 72.06 | 45.93 | 64.03 | 89.77 |
| 2 epochs | | | | | | | | |
|   Linear+SWA | 77.48 | 79.50 | 53.67 | 71.63 | **72.45** | 50.35 | 67.51 | 94.66 |
|   + Overlap | 70.08 | 60.23 | 39.08 | 66.63 | 60.93 | 48.06 | 57.50 | 80.62 |
|   SWA-only | 77.31 | 79.34 | 53.16 | 56.75 | 72.45 | 45.64 | 64.11 | 89.88 |
| 5 epochs | | | | | | | | |
|   Linear+SWA | **78.02** | 79.59 | 54.10 | **71.65** | 71.67 | 49.67 | 67.45 | 94.57 |
|   + Overlap | 68.82 | 58.50 | 38.14 | 66.24 | 58.88 | 46.99 | 56.26 | 78.88 |
|   SWA-only | 77.37 | 79.59 | 52.99 | 57.16 | 71.67 | 45.37 | 64.03 | 89.77 |

Table 14: Finetuning using SSD for a fixed dropout of 0.5 and a sliding window size schedule of 4, 8, 16, 32, 64.

## A.6 Further Ablations

### A.6.1 Sliding-window Size

In this ablation, we seek to investigate whether the observed collapse of the LA path in hybrid attention transfer methods exists across various sliding-window sizes. Due to the observed dominance of SWA with size $64$ and the short-context nature of the benchmarks used in our main experiments, as well as the literature, we investigate smaller sizes $\{8, 16, 32, 64\}$. We observe that, for both Mistral and Llama, the performance gap between LA+SWA and SWA-only shrinks as we increase the sliding-window size. In particular, for the smallest windows $\{8, 16\}$ during training and evaluation the hybrid model does benefit from adding the linear component, but this advantage steadily diminishes and becomes marginal once we reach SW $= 64$. Across all settings, LA-only and no attention remain almost identical, therefore showing no improvements in LA-only performance despite smaller window sizes. Furthermore, the observed gap between LA+SWA and SWA-only closes when evaluated with larger window sizes, which is not the case in our own methods, namely SSD. This indicates that, for all settings, the performance is still primarily attributed to SWA and that, as one would expect, the model increasingly ignores the additional LA path as the window size increases.

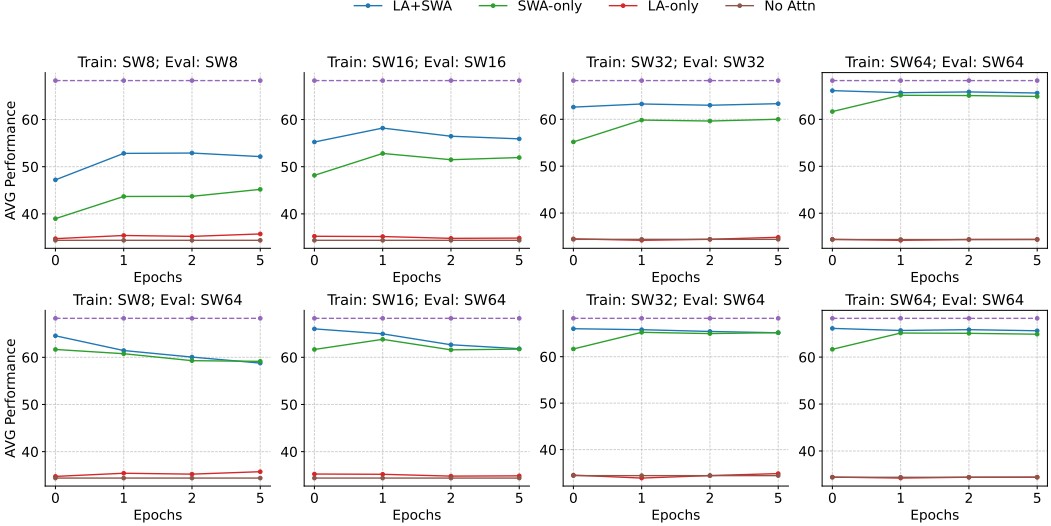

Figure 5: Measuring the performance of LoLCATs conversion for Mistral-v0.1 at various points during the conversion process for varying sliding-window size during training and evaluation.

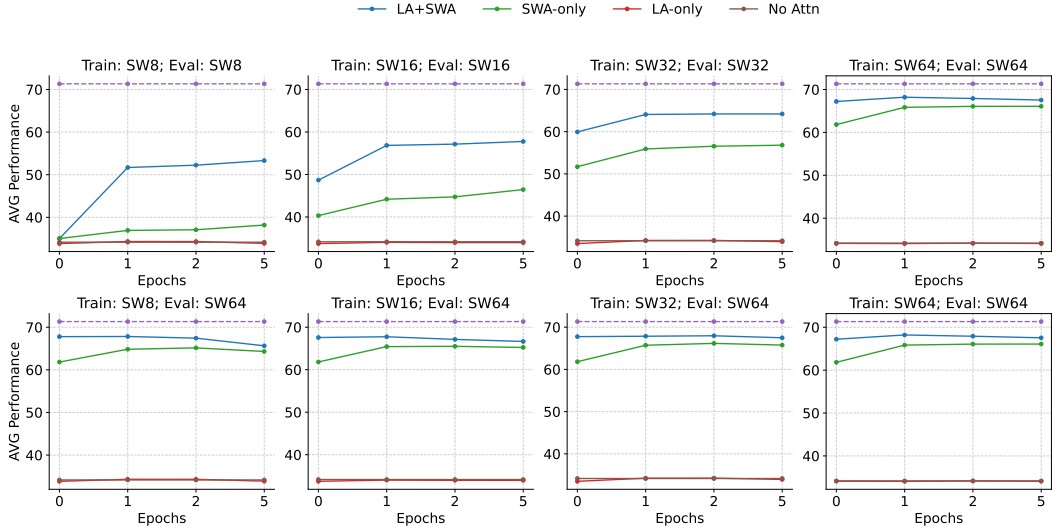

Figure 6: Measuring the performance of LoLCATs conversion for Llama3-8B at various points during the conversion process for varying sliding-window size during training and evaluation.

### A.6.2 MODEL SIZE

In this ablation, we seek to investigate whether the observed collapse of the LA path in hybrid attention transfer methods exists across various sliding-window sizes. In an effort to maintain a consistent architecture and pre-training method across model sizes, in order to isolate the effect of scaling the number of model parameters, we choose the Qwen2.5-Instruct (Yang et al., 2024a) model family. We chose these models as they provide a good diversity of model sizes within a range which stays mostly accessible to fine-tune and evaluate with a relatively low amount of computational resources (0.5B to 14B).

| Active Attn Modules | PIQA | ARC-E | ARC-C | HellaSwag | WG | MMLU | AVG | Rec. Perf |
|---|---|---|---|---|---|---|---|---|
| *Qwen2.5-0.5B-Instruct* | 70.40 | 65.40 | 33.79 | 52.44 | 56.35 | 45.71 | 54.02 | 100.00 |
| SWA + Linear | **70.29** | **67.21** | 34.81 | **45.96** | **57.62** | **34.09** | **51.66** | **95.65** |
| SWA only | 70.13 | 67.17 | **34.98** | 41.45 | **57.62** | 32.79 | 50.69 | 93.84 |
| Linear only | 58.49 | 38.26 | 22.87 | 27.74 | 49.17 | 22.94 | 36.58 | 67.72 |
| No Attention | 53.81 | 24.96 | 26.37 | 26.37 | 50.83 | 25.67 | 34.67 | 64.18 |
| *Qwen2.5-7B-Instruct* | 79.43 | 81.78 | 54.95 | 80.50 | 70.96 | 71.77 | 73.23 | 100.00 |
| SWA + Linear | 79.92 | 81.14 | 54.61 | 74.49 | **71.82** | 63.76 | 70.96 | 96.89 |
| SWA only | **80.14** | **81.69** | **55.03** | **76.18** | **71.82** | **64.25** | **71.52** | **97.66** |
| Linear only | 53.54 | 26.68 | 25.26 | 26.67 | 49.25 | 22.99 | 34.07 | 46.52 |
| No Attention | 52.29 | 26.47 | 27.05 | 25.49 | 48.38 | 22.95 | 33.77 | 46.12 |
| *Qwen2.5-14B-Instruct* | 81.45 | 85.73 | 62.37 | 84.35 | 75.77 | 78.93 | 78.10 | 100.00 |
| SWA + Linear | 81.12 | **85.27** | 61.18 | **78.36** | **76.09** | **69.63** | **75.28** | **96.38** |
| SWA only | **81.34** | 85.23 | **61.60** | 76.29 | **76.09** | 59.17 | 73.29 | 93.84 |
| Linear only | 52.07 | 26.52 | 24.83 | 27.93 | 48.62 | 23.06 | 33.84 | 43.33 |
| No Attention | 54.46 | 25.29 | 27.13 | 26.45 | 48.86 | 22.95 | 34.19 | 43.78 |

Table 15: Measuring the effect of attention components on benchmark accuracy for models trained with LoLCATs hybrid conversion (2 epochs of LoRA fine-tuning) across various model sizes in the Qwen2.5-Instruct model family.

### A.6.3 BASE MODELS

Our main experiments make use of instruction-tuned models. In this section, we repeat a subset of such experiments on the corresponding base models in order to determine whether instruction-tuning affects the results. Table 16 confirms that the base checkpoints also suffer from a collapse of LA and

dominance of SWA, as seen in Table 1. Figure 7 shows very similar improvements in LA utilisation and larger gap between SWA-only and inference-time hybrid performance as Figure 2. It therefore appears as though instruction-tuning has no impact on our findings.

| Active Attn Modules | PIQA | ARC-E | ARC-C | HellaSwag | WG | MMLU | AVG | Rec. Perf |
|---|---|---|---|---|---|---|---|---|
| *Mistral-7B* | 80.79 | 80.81 | 54.10 | 81.05 | 73.88 | 59.55 | 71.70 | 100.00 |
| SWA + Linear | **80.41** | 81.27 | **52.65** | **76.66** | **74.11** | **48.77** | **68.98** | **96.21** |
| SWA only | 80.30 | **81.31** | **52.65** | 73.55 | **74.11** | 44.28 | 67.70 | 94.43 |
| Linear only | 53.32 | 29.34 | 24.74 | 27.44 | 47.59 | 25.88 | 34.72 | 48.42 |
| No Attention | 52.88 | 26.52 | 25.09 | 25.85 | 49.57 | 22.95 | 33.81 | 47.16 |
| *Llama3-8B* | 79.49 | 80.18 | 53.41 | 79.26 | 72.85 | 62.03 | 71.20 | 100.00 |
| SWA + Linear | **79.43** | 79.92 | 52.13 | **73.26** | **73.72** | **47.54** | **67.67** | **95.03** |
| SWA only | 78.94 | **80.05** | **52.65** | 66.54 | **73.72** | 45.68 | 66.26 | 93.06 |
| Linear only | 52.50 | 25.97 | 25.94 | 26.62 | 48.93 | 22.95 | 33.82 | 47.50 |
| No Attention | 54.19 | 26.22 | 24.23 | 25.46 | 50.59 | 22.95 | 33.94 | 47.67 |
| *Llama3.1-8B* | 79.98 | 81.61 | 53.41 | 79.02 | 73.24 | 63.32 | 71.76 | 100.00 |
| SWA + Linear | **79.49** | 80.93 | **53.41** | **72.47** | **73.64** | 45.60 | **67.59** | **94.18** |
| SWA only | 79.33 | **81.02** | **53.41** | 66.73 | **73.64** | **46.40** | 66.76 | 93.02 |
| Linear only | 53.54 | 27.40 | 22.95 | 26.40 | 51.62 | 22.95 | 34.14 | 47.58 |
| No Attention | 54.41 | 26.68 | 24.66 | 25.93 | 49.17 | 22.95 | 33.97 | 47.33 |

Table 16: Measuring the effect of attention components on benchmark accuracy for models trained with LoLCATs hybrid conversion (2 epochs of LoRA fine-tuning) for base models (not instruction-tuned).

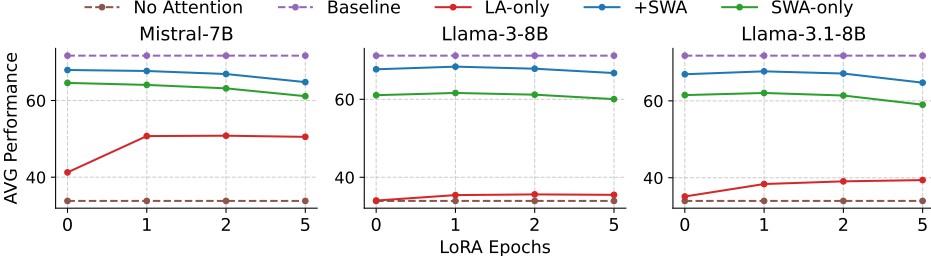

Figure 7: Adding SWA at inference time at various points of the conversion process (weights transfer + LoRA fine-tuning). LoRA refers to LA-only finetuning, with epoch 0 corresponding to weights transfer only.

### A.6.4 INFERENCE-TIME HYBRID WITH ATTENTION OUTPUTS TRANSFER

In this ablation, we apply LA-only fine-tuning after applying the attention outputs transfer objective, as ablated in Table 9. Interestingly, while it makes for lower LA utilisation in Mistral and Llama-3.1, Llama-3 appears to do much better with the transfer objective than with weights transfer (Figure 7).

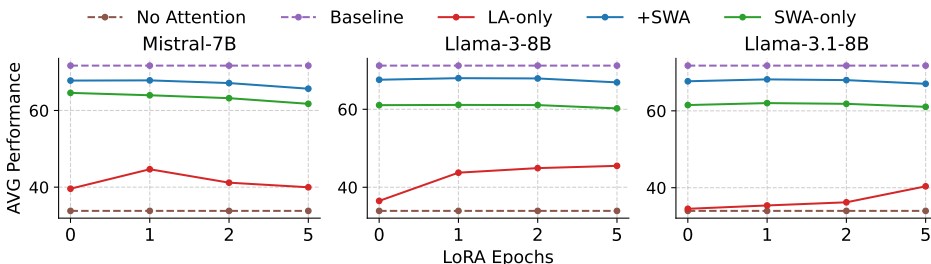

Figure 8: Adding SWA at inference time at various points of the conversion process (attention outputs transfer + LoRA fine-tuning). LoRA refers to LA-only finetuning, with epoch 0 corresponding to attention transfer only.

### A.6.5   HEDGEHOG WITH FULL-PARAMETER FINE-TUNING

As mentioned in Section 3.3, HedgeHog also makes use of full-parameter fine-tuning in settings outside of pre-trained conversion, where it uses LoRA. For completeness, we investigate the performance of HedgeHog with full-parameter fine-tuning in Table 17. We find that results follow that of the more targeted LoRA fine-tuning (Table 8), although suggests a slight decrease in downstream performance.

| Active Attn Modules | PIQA | ARC-E | ARC-C | HellaSwag | WG | MMLU | AVG | Rec. Perf |
|---|---|---|---|---|---|---|---|---|
| *Mistral-7B-Instruct* | 79.27 | 80.01 | 52.22 | 74.60 | 69.93 | 53.51 | 68.26 | 100.00 |
| No Attention | 53.92 | 25.63 | 24.66 | 25.99 | 50.67 | **25.51** | 34.40 | 50.39 |
| LA Weights Transfer | 68.23 | 63.22 | 30.46 | 47.21 | **52.17** | 23.96 | 47.54 | 69.65 |
| +Full FT 1 Epoch | **71.22** | 67.51 | 35.58 | 58.11 | 51.93 | **26.11** | 51.74 | 75.81 |
| +Full FT 2 Epochs | 71.16 | **67.85** | **35.75** | **58.47** | 51.54 | 25.92 | **51.78** | **75.86** |
| *Llama3-8B-Instruct* | 78.13 | 81.69 | 56.66 | 75.94 | 71.67 | 63.85 | 71.32 | 100.00 |
| No Attention | **55.17** | 26.73 | 22.87 | 26.13 | 51.14 | 22.95 | 34.17 | 47.90 |
| LA Weights Transfer | 53.26 | 28.03 | 23.55 | 26.62 | 49.80 | 23.23 | 34.08 | 47.78 |
| +Full FT 1 Epoch | **55.17** | **28.41** | **24.91** | **26.78** | 50.43 | 23.43 | 34.86 | 48.87 |
| +Full FT 2 Epochs | 54.73 | 28.07 | 24.57 | 26.65 | **52.17** | **23.51** | **34.95** | **49.00** |
| *Llama3.1-8B-Instruct* | 80.14 | 81.82 | 55.20 | 79.14 | 73.72 | 68.05 | 73.01 | 100.00 |
| No Attention | 53.59 | 26.89 | **25.94** | 26.19 | 49.09 | 22.95 | 34.11 | 46.72 |
| LA Weights Transfer | 55.06 | 33.63 | 21.59 | 27.84 | 49.25 | **25.00** | 35.40 | 48.48 |
| +Full FT 1 Epoch | **60.07** | **42.42** | 23.46 | 29.39 | **51.85** | 23.76 | **38.49** | **52.72** |
| +Full FT 2 Epochs | **60.07** | 42.26 | 22.95 | **29.49** | 51.38 | 23.83 | 38.33 | 52.50 |

Table 17: Measuring the performance of HedgeHog conversion with full parameter finetuning instead of targeted LoRA at various points during the conversion process. Full FT refers to a LA-only finetuning of all base model parameters.

### A.6.6   LONG-CONTEXT EVALUATION

As mentioned in Section 1, LA is often motivated as an efficient way to process long sequences due to its linear complexity. However, as discussed in Appendix A.3.1, vanilla LA, as employed in this work, has well-documented limitations for long-context tasks. Nevertheless, for completeness, we present our main diagnostics on the SCROLLS (Shaham et al., 2022) and LongBench (Bai et al., 2024) benchmarks (Figs.9 and10, respectively). As anticipated, the resulting models fail to recover the performance of the base models. More importantly, the results exhibit high variability across models, methods, and benchmarks, with some configurations even underperforming the no-attention baseline. Consequently, these experiments do not yield actionable insights for the hybrid conversion methods that are the focus of this work. Future work should evaluate long-context performance using hybrid conversion models that incorporate LA variants explicitly designed for extended sequences, such as those with learned mixing terms, state decay mechanisms, and gated updates, combined with training on longer sequences and adaptation strategies for unseen position embeddings.

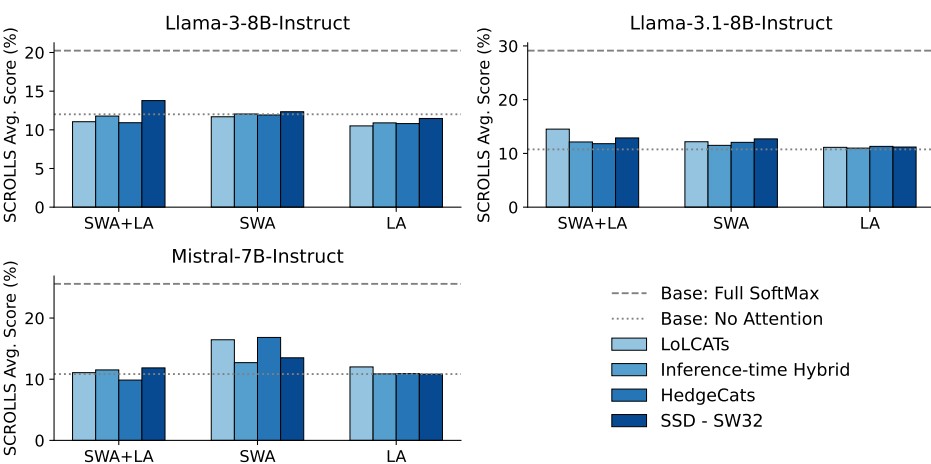

Figure 9: Comparing performance of each hybrid attention component on the SCROLLS benchmark for all conversion methods. All methods are fine-tuned for 2 epochs. "SSD - SW32" refers to an SSD model converted using a fixed sliding-window size of 32 and scheduled dropout of 0.9, 0.75 across fine-tuning epochs.

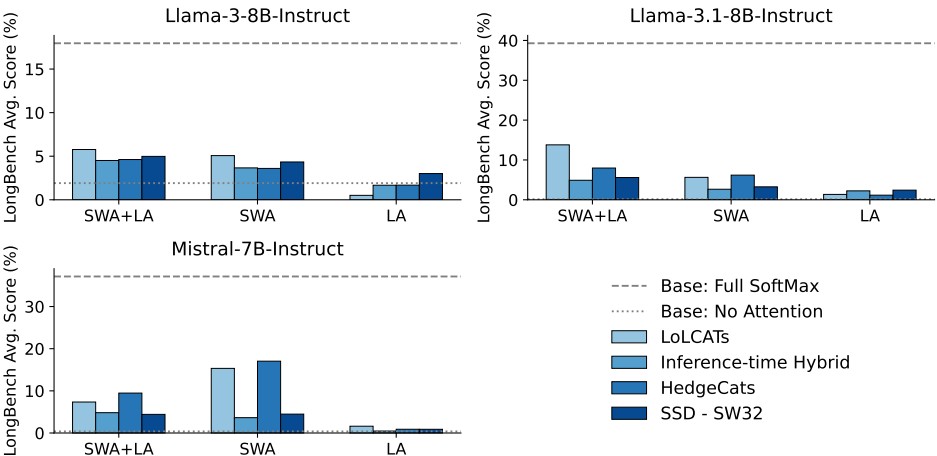

Figure 10: Comparing performance of each hybrid attention component on the SCROLLS benchmark for all conversion methods. All methods are fine-tuned for 2 epochs. "SSD - SW32" refers to an SSD model converted using a fixed sliding-window size of 32 and scheduled dropout of 0.9, 0.75 across fine-tuning epochs.

