# OpenReview forum: "Paying Attention to Hybrid Attention: Untangling the Issues with Conversion Methods"
_ICLR.cc/2026/Conference — Submitted to ICLR 2026_

### Official Review · Reviewer_ry4B · 2025-10-27

**Soundness:** 2
**Presentation:** 3
**Contribution:** 3
**Rating:** 4
**Confidence:** 4

**Summary:**

This paper studies hybrid attention conversion for LLMs. That is, take a pretrained softmax Transformer and convert it into a hybrid that combines linear attention (LA) with sliding-window attention (SWA) to achieve linear-time long-context inference. The authors’ key finding is that, with existing conversion/training practices, hybrids often ignore the linear branch and rely almost entirely on SWA, so reported “hybrid” gains are misleading. They provide component-level diagnosis and propose three remedies that make hybrids use LA while preserving efficiency: inference-time hybrid addition of SWA to LA-only conversions; HedgeCATs, which pairs attention-weight transfer with brief LoRA fine-tuning; and Scheduled Sliding-window Dropout (SSD), which randomly mask out the SWA branch during fine-tuning via a dropout schedule. These remedies recover much of the base model’s accuracy and ensure the linear path is actually used.

**Strengths:**

1, Identify a subtle but important pitfall that hybrid attention works without ever using LA; the paper provides component-level diagnostics and a training-time fix (SSD) to prevent the issue.
2, The paper conducts thorough ablations between swa-only VS hybrid, LA only VS attention sinks VS no attention.
3, The proposed fixes (like disabling sliding-window attention during parts of training) are simple to implement and work with existing hybrid models, improving linear attention performance without needing major architectural changes.
4, The paper does a nice job explaining how linear attention works and why it should theoretically help. It also presents detailed experiments that make the issue easy to understand.

**Weaknesses:**

1, The experiments focus on tasks that don’t actually require long-range reasoning. Since the whole point of linear attention is handling long contexts, it would be more convincing to test on tasks where long-distance dependencies really matter.
2, The paper shows that linear attention contributes more after their training fixes — but they don’t define a clear metric for what counts as “meaningful use” of LA. A stronger definition would help future work evaluate hybrids more fairly.
3, The new training methods (e.g. SSD, HedgeCATs) may add extra computational cost. The paper doesn’t provide expositions of how much extra compute or latency they introduce.
4, The success of the SSD schedule may depend on how it is tuned (like dropout rate and window size), but the paper doesn’t provide enough guidance on how to set these values for different models.
5, Even after the fixes, linear attention alone still performs poorly. The paper touches on possible reasons but doesn’t give conclusive evidence on what fundamentally limits LA’s quality.

**Questions:**

1, Can you include at least one true long-context task? For example: document-level QA, long-context retrieval, needle in haystack, or benchmarks for long-range reasoning.
2, How do we measure LA’s contribution more formally?
3, What is the exact compute and latency impact? How much extra fine-tuning time does SSD or HedgeCATs require compared to standard hybrid conversion? And does inference get slower once LA is finally used?
4, What SSD settings should others start with? Do you have rule-of-thumb recommendations so practitioners can reproduce results without extensive tuning?
5, What do you think is the real bottleneck in linear attention? Is it the feature map design, insufficient hidden dimension, optimization challenges, or something else? Any general advice for making LA itself stronger?

---

> ### Author Response · Authors · 2025-11-26
>
> > The experiments focus on tasks that don’t actually require long-range reasoning. Since the whole point of linear attention is handling long contexts, it would be more convincing to test on tasks where long-distance dependencies really matter
>
> > Can you include at least one true long-context task? For example: document-level QA, long-context retrieval, needle in haystack, or benchmarks for long-range reasoning.
>
> We are still working on identifying an appropriate long-context benchmark. Although we have tested several options, including the reviewer’s suggestion, we consistently observe collapse of the linear component, irrespective of the conversion method used. While the reviewer is correct that long-context settings stand to benefit most from linear attention, to the best of our knowledge the field has yet to demonstrate an implementation of linear attention (without additional gating mechanisms) that achieves competitive performance in this regime. Because our work focuses specifically on the collapse of linear attention under hybrid conversion methods (using 1024-token sequences), existing long-context benchmarks may not be directly aligned with our setting. Our Discussion section mentions this limitation and suggests that a promising direction would be to expand conversion methods to gated formulations of linear attention.
>
> Nonetheless, we are continuing to run longer-context evaluations in the hope of identifying a benchmark that yields more informative results. We have also softened the wording in Section 1 to better reflect this gap between motivation and current performance.
>
> > The paper shows that linear attention contributes more after their training fixes — but they don’t define a clear metric for what counts as “meaningful use” of LA. A stronger definition would help future work evaluate hybrids more fairly.
>
> > How do we measure LA’s contribution more formally?
>
> We agree with the reviewer that a single quantifiable metric would be most convenient for future use. Despite our efforts in analysis to this end, we have so far found no other metrics that are seemingly meaningful or consistent with the clear component-wise gaps in performance that make up our main results.
>
> For example, reviewer nPh2 pertinently suggested looking at the share of attention weights between LA and SWA. However, this would require both SWA and LA to share a common denominator and an equal or constant mixing term in order to make any kind of comparison between the two components. While LoLCATs did do this in their implementation, as well as *learning* a mixing term, we have abstained from either of these options in order to minimise the model’s ability to diminish LA outputs simply by decreasing the magnitude of LA’s raw attention weights when learning projection $\phi$. We mentioned this briefly in section 3.1. The motivation behind this choice was to fully isolate other factors, namely the transfer objective, to investigate their involvement in this observed collapse, since both our implementation and LoLCATs’ see this LA collapse despite these different normalisation strategies. That being said, we have investigated the magnitude of raw attention weights in LoLCATs’ codebase and do consistently observe a significantly smaller, but non-negligible denominator for the LA component.
>
> Furthermore, in order to find a more implementation-agnostic metric and further explain the collapse of LA, we have investigated several other potential metrics. The magnitude of attention outputs appeared to be an interesting direction as attending to lower-magnitude values (namely the attention sinks) may decrease the LA path’s contributions to the hybrid output, although, so far, while there is a notable difference in magnitude, it is not quite large enough to explain the full collapse of LA observed in LoLCATs. (2) We are also looking at differences in attention weights and their entropy, as well as the similarity of outputs between LA and softmax attention applied to the corresponding context. While we also observe an imbalance here, it is once again not large enough to explain the observed collapse. While, together, these analyses are likely to explain the observed performance gaps, individually they only appear to show part of the picture, and are therefore weak candidates for a robust unified metric.
>
> However,  we are happy to add such analysis to the appendix if the reviewer believes this would be a valuable addition. In the meantime, we have also clarified that we recommend quantifying the gaps in performance between hybrid and SWA-only, as well as no attention and LA-only, as the main thing to check for successful conversions of pre-trained transformers, and formalised this metric in section 5.
>
> If the reviewer feels that any further changes to the text are needed to better reflect the points above, we will be happy to accommodate.

---

> ### Author Response · Authors · 2025-11-26
>
> > The new training methods (e.g. SSD, HedgeCATs) may add extra computational cost. The paper doesn’t provide expositions of how much extra compute or latency they introduce.
>
> > What is the exact compute and latency impact? How much extra fine-tuning time does SSD or HedgeCATs require compared to standard hybrid conversion? And does inference get slower once LA is finally used?
>
> We thank the reviewer for raising this important point regarding scalability. We appreciate the opportunity to clarify the computational profile of our proposed methods compared to LoLCATs.
>
> Regarding our proposed methods, they all incur the same inference-time computational cost as LoLCATs models. During training, our methods have the same theoretical computational training cost as LoLCATs. For attention transfer, while weight transfer causes the linear component to incur quadratic complexity during conversion, this does not affect the asymptotic $\mathcal{O}$ time complexity as the target softmax attention output already incurs quadratic cost. However, in practice, it does result in slightly increased wall-clock time over the attention output transfer objective. As for the fine-tuning stage, LA-only fine-tuning used with inference-time hybrids reduces to $\mathcal{O}(nd^2)$, while SSD holds the same complexity as LoLCATs but may see lower wall-clock times in the case of smaller, scheduled sliding-windows. Finally, HedgeCATs does not change this stage.
>
> We have added a section detailing this to the appendix (Section A.4) in our revised manuscript.
>
> > The success of the SSD schedule may depend on how it is tuned (like dropout rate and window size), but the paper doesn’t provide enough guidance on how to set these values for different models.
>
> > What SSD settings should others start with? Do you have rule-of-thumb recommendations so practitioners can reproduce results without extensive tuning?
>
> We thank the reviewer for pointing out this weakness. We agree that in order to make our method practical for other practitioners we should provide some guidance as to how the schedule hyperparameters should be decided. To this end, we have provided further motivation as per our choices in the experiments presented (section 4.3, first paragraph). The key reason being that the model should initially have to rely more on LA outputs, then gradually gain access to more SWA context and outputs.
>
> In the meantime, following reviewer Eoas’s comment, we are ablating schedules. We will update the manuscript with our findings and include a dedicated section on hyperparameter recommendations.
>
> However, we are unlikely to be able to complete a full hyper-parameter search for all models before the end of the rebuttal period, given the large search space and our limited computational resources which are already spread thin between other additional experiments. To this end, as we fully agree with the reviewer, we explicitly mention that this hyper-parameter tuning is neccessaruy and costly, and present this as a weakness to the SSD method, which the other two methods lack, in the Discussion section (Limitations and Future Work paragraph).

---

> > ### Author Response · Authors · 2025-11-26
> >
> > > Even after the fixes, linear attention alone still performs poorly. The paper touches on possible reasons but doesn’t give conclusive evidence on what fundamentally limits LA’s quality.
> >
> > > What do you think is the real bottleneck in linear attention? Is it the feature map design, insufficient hidden dimension, optimization challenges, or something else? Any general advice for making LA itself stronger?
> >
> > We agree that understanding what *fundamentally* limits the quality of linear attention (LA) is an important question. However, we consider a complete answer to this question to be beyond the scope of our work, and is in fact the focus of a growing, separate sub-area of research.
> >
> > On top of the limitations already mentioned in our main text (namely section 5, Limitations and Future Work), several other papers explicitly analyse structural and optimisation limitations of LA. For example, Qin et al. (2022) [1] show that kernel-based linear Transformers can suffer from training instability, and attention dilution. The latter can also be attributed to higher-entropy, hence low-spikyness attention scores, which motivate the use of HedgeHog [2]. Fan et al. (2024) [3] attribute much of the performance gap between linear and softmax attention to the intrinsically low-rank nature of standard linear attention feature maps, which limits expressivity.  Arora et al. (2024) [4] show that simple linear-attention language models lie on a recall–throughput Pareto frontier: fixed-size state architectures (including LA and state-space models) systematically trade off recall quality against efficiency, and struggle to match full softmax attention recall without significantly increasing state size. A separate line of work also performs broad empirical and theoretical analyses of hybrid linear attention architectures and documents persistent quality gaps and trade-offs across variants [5].
> >
> > Our contribution is orthogonal to these efforts: we empirically study the collapse of the linear component *within hybrid attention conversions* under a fixed set of kernels and conversion procedures, rather than attempting to resolve the broader open question of what fundamentally limits LA across architectures and training regimes. We therefore see a full explanation of LA’s intrinsic limitations as an important but distinct research direction.
> >
> > [1] Z. Qin et al., *“The Devil in Linear Transformer,”* EMNLP 2022
> >
> > [2] M. Zhang et al., “*The hedgehog & the
> > porcupine: Expressive linear attentions with softmax mimicry”,*
> > ICLR 2024
> >
> > [3] Q. Fan et al., *“Breaking the Low-Rank Dilemma of Linear Attention,”* CVPR 2025
> >
> > [4] S. Arora et al., *“Simple Linear Attention Language Models Balance the Recall–Throughput Tradeoff,”* 2024
> >
> > [5] D. Wang et al., *“A Systematic Analysis of Hybrid Linear Attention,”* 2025

---

> > > ### Comment · Reviewer_ry4B · 2025-11-27
> > > **Response to authors**
> > >
> > > I thank the authors for the rebuttal. I appreciated the clarifications on the training procedures, the discussion of compute considerations, and the explanations around SSD and HedgeCATs. The citations of known limitations of linear attention and the discussion of alternative measures of LA usage are also helpful.
> > >
> > > That said, several of my core concerns remain insufficiently addressed or unaddressed. In particular, the paper still lacks evaluation on any true long-context task, a concrete or quantitative metric for “meaningful use” of the linear branch, guidance on SSD hyperparameters, and a clearer picture of the compute/latency tradeoffs. These elements are important for assessing the practical impact and generality of the work. While the authors’ responses provide helpful insights, they do not materially fill the empirical or methodological gaps in the submission.
> > >
> > > My original score therefore remains unchanged.

---

> > > > ### Author Response · Authors · 2025-12-03
> > > >
> > > > We thank the reviewer for their prompt response. However, all the points raised by the reviewer as still unaddressed or insufficiently addressed have been substantially addressed either in our responses, the updated text, or both. The reviewer provides no details as to how our detailed changes fail to address their concerns.
> > > >
> > > > More specifically, we would like the reviewer to clarify how the following existing changes do not satisfy their concerns:
> > > >
> > > > 1. “a concrete or quantitative metric for “meaningful use” of the linear branch”
> > > >     * We have discussed such interpretations quantitatively throughout the paper, and provided Equation 5 in Section 5 as a quantitative metric to this end. Furthermore, in our response, we outlined other candidates that we considered, as well as their shortcomings, and offered to add this to the Appendix, which the reviewer did not address.
> > > > 2. “guidance on SSD hyperparameters”
> > > >     * We have further clarified our hyper-parameter choices to guide future works (Section 4.3), as the reviewer initially asked “What SSD settings should others start with?”, as well as explicitly acknowledging that the expensive hyper-parameter tuning, which may be required for SSD, is a clear weakness for this approach compared to our other proposed methods (Section 5).
> > > > 3. “a clearer picture of the compute/latency tradeoffs”
> > > >     * We have added Appendix A.4 which presents and discusses both theoretical and empirically-measured efficiency trade-offs, answering all questions and points initially raised by the reviewer.
> > > > 4. “evaluation on any true long-context task”
> > > >     * We have clearly justified the reasons for abstaining from presenting long-context evaluations in our initial response, which the reviewer has failed to address in any way. To this end, we have added to the extended related works section in Appendix A.3.1, specifically discussing linear models in a long-context setting to further illustrate why evaluating standard linear attention-based hybrids, which is the focus of this work, does not make sense in our case. However, for completeness, we have also evaluated all of our models and conversion methods on both the SCROLLS and LongBench benchmarks, repeating our diagnostics presented in our main experiments. These new results are now available in Appendix A.6.6, but expectedly exhibit high variability and lack meaningful signal.

---

### Official Review · Reviewer_8pxg · 2025-11-01

**Soundness:** 3
**Presentation:** 3
**Contribution:** 3
**Rating:** 6
**Confidence:** 4

**Summary:**

This work addresses an issue in hybrid attention architectures that combine linear attention (LA) and sliding-window softmax (SWA). It identifies a flaw in current post-training linearisation methods, where models often rely almost entirely on the SWA component at the expense of the linear attention pathway, leading to misleading performance attributions. Through component-level diagnostics, the authors expose this behavior, which stems from insufficient evaluation methods on benchmarks.

The authors propose three remedies: (1) a zero-shot inference-time hybridisation of linear-only approaches with SWA, (2) HedgeCATs, integrating attention-weight transfer with targeted LoRA fine-tuning, and (3) Scheduled Sliding-window Dropout (SSD) to suppress softmax reliance during training. These solutions ensure balanced usage of attention pathways, maintain computational efficiency, and recover most of the base model's performance while preserving the integrity of linear attention adoption in hybrid conversions.

**Strengths:**

* **Novel contribution & clear problem formulation**: The authors identify a previously unknown issue in hybrid attention conversions, which I find quite interesting. The paper clearly articulates the problem of existing hybrid attention conversions bypassing the linear component and relying on SWA.
* **Well-motivated solutions & ablations**: The authors propose three practical solutions to address the issue, which are well-motivated and based on a thorough understanding of the problem. The paper provides component-level diagnostics to quantify each component's contribution, making it possible to detect and address imbalances.
* **Experimental evaluation & Results**: The authors provide multiple evaluations to demonstrate the efffectiveness of their method and the results are strong!
* **Clear writing style**: The paper's writing style is clear, concise, and easy to follow, making it accessible to a broad audience.

**Weaknesses:**

* The main goal of linear attention and its hybrid cousins is to achieve better efficiency i.e memory, throughput or even latency. However this work only provides quality benchmark numbers. In my experience such algorithmic improvements like the one in this paper do not necessarily imply better wall clock speed, memory or throughput. Could the authors provide more details on how their methods impact peformance along these axes ?
* Comment on Line 80: Standard Softmax attention does not necessarily need to incur O(T^2) memory. Depending on the implementation it could incur only O(T) memory if you use the Softmax trick.

**Questions:**

* One of the main contributions of LolCATS is to propose a method that can scale for linear attention to very large models i.e up to LLAMA 405B. How does the proposed method in this paper scale compared to LolCATS ?

---

> ### Author Response · Authors · 2025-11-26
>
> > The main goal of linear attention and its hybrid cousins is to achieve better efficiency i.e memory, throughput or even latency. However this work only provides quality benchmark numbers. In my experience such algorithmic improvements like the one in this paper do not necessarily imply better wall clock speed, memory or throughput. Could the authors provide more details on how their methods impact peformance along these axes ?
>
> This is an important point, which was also raised by Reviewer Eoas, and we thank you for highlighting it. We originally did not report efficiency metrics because our implementation is not optimised for maximum efficiency (which would likely require a custom, kernel-level implementation) and therefore does not reflect the theoretical advantages when compared to flash-attention; instead, our primary focus was on faithfully replicating and improving converted model quality. Nonetheless, we agree that it is important to assess and report performance along efficiency axes for completeness. In the revised manuscript, we therefore include benchmarks comparing our hybrid models with the full softmax baseline in terms of efficiency, and we have added a new table with these empirical results, together with a theoretical O(⋅) complexity analysis. These additions to the paper can be found  section A.4 of the appendix.
>
> > Comment on Line 80: Standard Softmax attention does not necessarily need to incur O(T^2) memory. Depending on the implementation it could incur only O(T) memory if you use the Softmax trick.
>
> The reviewer appears to be pertinently referring to the “softmax trick”, which is the numerically-stable online softmax used in memory-efficient attention kernels such as FlashAttention [1].
>
> While this technique reduces peak activation memory from $\mathcal{O}(T^2)$ to $\mathcal{O}(T)$ by not materialising the full attention matrix, it does not reduce the computational complexity, which remains $\mathcal{O}(T^2)$ because all $QK^T$ dot products must still be computed.
>
> We have updated the line in question to clarify this. We thank the reviewer for their comment.
>
> [1] Tri Dao et al., “*FlashAttention: Fast and Memory-Efficient Exact Attention with IO-Awareness”*, NeurIPS 2022
>
> > One of the main contributions of LolCATS is to propose a method that can scale for linear attention to very large models i.e up to LLAMA 405B. How does the proposed method in this paper scale compared to LolCATS ?
>
> We thank the reviewer for raising this question regarding scalability. We would like to clarify how our methods compare to the engineering strategies used in LoLCATs. LoLCATs scales up to 405B models primarily through sequential conversions of chunked layers, processing the model in chunks for memory efficiency and training stability. This approach is orthogonal to the specific conversion method used. Therefore, while our current codebase does not implement this specific scaling strategy, our methods are inherently compatible with the LoLCATs sharding logic, and should thus benefit from the same scalability profile as the original pipeline. Unfortunately, we do not have access to the necessary computational resources to empirically verify this ourselves at this time.

---

### Official Review · Reviewer_nPh2 · 2025-11-01

**Soundness:** 2
**Presentation:** 3
**Contribution:** 2
**Rating:** 4
**Confidence:** 4

**Summary:**

This paper studies post-training hybrid linearization of Transformers that mix linear attention (LA) with sliding-window softmax attention (SWA). The authors identify a systematic failure mode: under common hybrid training procedures, the model collapses to using SWA almost exclusively, with the LA branch effectively ignored. They demonstrate this via component-level diagnostics (SWA-only vs LA-only vs combined) on Mistral-7B and Llama-3/3.1-8B (including LoLCATs checkpoints). Through controlled ablations, they attribute the collapse primarily to the hybrid attention-output transfer objective and reduced feature-map dimensionality; they also confirm exponential-family activations outperform ReLU/ELU. They propose three remedies: (i) an inference-time hybridization (zero-shot SWA addition to LA-only conversions), (ii) HedgeCATs (HedgeHog-style attention-weight transfer followed by short LoRA fine-tuning with early stopping), and (iii) Scheduled Sliding-window Dropout (SSD), which stochastically suppresses SWA during fine-tuning. These interventions recover much of the base model accuracy while ensuring measurable LA usage, addressing misleading attribution in hybrid conversions.

**Strengths:**

- **Diagnosis of a failure mode**: The authors show various component-wise ablations that surface how hybrid conversion performance can effectively be preserved with just SWA, with little ostensible contribution from linear attention dependence.
- **Careful component-level analysis**: The paper isolates the role of the hybrid transfer objective, feature-map dimensionality, and activation choice (exponential-family vs ReLU/ELU), offering actionable guidance.
- **Remedies are simple and practical**: The authors propose simple solutions such as inference-time hybridization, HedgeCATs (weight-transfer + short LoRA with early stopping), and SSD (structured dropout on SWA) that demonstrably increase LA utilization without large overhead.

**Weaknesses:**

## **Potentially insufficient evaluation**

While the evaluated tasks are consistent with prior works [1] and I appreciated the comprehensive component ablation considering base LLM, linear attn feature map, and conversion objective, I think these tasks aren't actually the best for showing over-reliance on SWA.

**Evaluation Task**.

Many of these tasks are short-context LM-Eval tasks, so it makes sense that performance between SWA and SWA + Linear may be comparable.
 - Did the authors study other long-context retrieval tasks? Even with simple ones such as (multi-query) associative recall or passkey retrieval, showing that SAW and SWA + Linear here get the same performance would make the diagnosis much more compelling (i.e., if we're in a regime where the model has to attend 8k tokens back, and the sliding window size is only 32 (outside the receptive field), and SWA + Linear does no better than SWA alone, then this would present compelling evidence.
   - I noticed in the LoLCATs paper that Table 21 describes some related results here in passkey retrieval [1], perhaps the authors can look into this during the rebuttal period?
- It would also be interesting to see the effective of ablating softmax attention window size in the initial diagnostic (not just in the scheduling method)

**Diagnosis metric**.

In another vein, just the end-task performance alone may not be granular enough to suggest the linear attention layers are not contributing anything. Did the authors consider visualizing the attention weights?
- For example, again if the SWA + Linear attention weights show no activation in the linear attention weights, this would also present more compelling evidence.
- The authors could provide a quantifiable metric here too, i.e., e.g., reporting how much weight is outside the SWA size.

[1] Michael Zhang, Simran Arora, Rahul Chalamala, Alan Wu, Benjamin Spector, Aaryan Singhal,
Krithik Ramesh, and Christopher R´ e. Lolcats: On low-rank linearizing of large language models.
arXiv preprint arXiv:2410.10254, 2024a.

**Questions:**

Please see the questions raised in the weaknesses section above.

---

> ### Author Response · Authors · 2025-11-26
>
> > Did the authors study other long-context retrieval tasks? Even with simple ones such as (multi-query) associative recall or passkey retrieval, showing that SAW and SWA + Linear here get the same performance would make the diagnosis much more compelling (i.e., if we're in a regime where the model has to attend 8k tokens back, and the sliding window size is only 32 (outside the receptive field), and SWA + Linear does no better than SWA alone, then this would present compelling evidence.
> > I noticed in the LoLCATs paper that Table 21 describes some related results here in passkey retrieval [1], perhaps the authors can look into this during the rebuttal period?
>
> We are still working on identifying an appropriate long-context benchmark. Although we have tested several options, including the reviewer’s suggestion, both SWA and LA, individually and combined, fail to recover any meaningful proportion of the base model’s performance in such a setting, irrespective of the conversion method used. To the best of our knowledge the field has yet to demonstrate an implementation of linear attention (without additional gating mechanisms, or task-specific fine-tuning) that achieves competitive performance in this regime, especially for sparse recall tasks. For example, LoLCATs’ results on passkey retrieval shows 0% accuracy without specifically converting the underlying pre-trained model exclusively on such examples. This likely yields a learned $\phi$ which exclusively focuses on returning the passkey and is therefore meaningless for general pre-trained conversion. Our initial results on this benchmark are consistent with this.
>
> Therefore, as our work focuses specifically on the collapse of linear attention under general, pre-trained hybrid conversion methods using task-agnostic 1024-token sequences, long-context retrieval benchmarks may not be directly aligned with our setting. Our Discussion section mentions this limitation within long-context in general and suggests that a promising direction would be to expand conversion methods to gated formulations of linear attention, which have shown more promising performance in long-context settings.
>
> > It would also be interesting to see the effective of ablating softmax attention window size in the initial diagnostic (not just in the scheduling method)
>
> We thank the reviewer for this pertinent suggestion. In response, we ran an ablation converting and evaluating pre-trained models using different sliding-window sizes (8,16,32,64) for both Mistral and Llama using the LoLCATs conversion method. We have added these results to the revised manuscript in the appendix (Section A.6.1).

---

> > ### Author Response · Authors · 2025-11-26
> >
> > > In another vein, just the end-task performance alone may not be granular enough to suggest the linear attention layers are not contributing anything. Did the authors consider visualizing the attention weights? For example, again if the SWA + Linear attention weights show no activation in the linear attention weights, this would also present more compelling evidence.
> > > The authors could provide a quantifiable metric here too, i.e., e.g., reporting how much weight is outside the SWA size
> >
> > We agree with the reviewer that a single quantifiable metric would be most convenient for future use. Despite our efforts in analysis to this end, we have so far found no other metrics that are seemingly meaningful or consistent with the clear component-wise gaps in performance that make up our main results.
> >
> > We agree that monitoring the share of attention weights across such components is a pertinent and likely meaningful metric. However, this would require both SWA and LA to share a common denominator and an equal or constant mixing term in order to make any kind of comparison between the two components. While LoLCATs did do this in their implementation, as well as *learning* a mixing term, we have abstained from either of these options in order to minimise the model’s ability to diminish LA outputs simply by decreasing the magnitude of LA’s raw attention weights when learning projection \phi. We mentioned this briefly in section 3.1. The motivation behind this choice was to fully isolate other factors, namely the transfer objective, to investigate their involvement in this observed collapse, since both our implementation and LoLCATs’ see this LA collapse despite these different normalisation strategies. That being said, we have investigated the magnitude of raw attention weights in LoLCATs’ codebase and do consistently observe a significantly smaller, but non-negligible denominator for the LA component.
> >
> > Furthermore, in order to find a more implementation-agnostic metric and further explain the collapse of LA, we have investigated several other potential metrics. The magnitude of attention outputs appeared to be an interesting direction as attending to lower-magnitude values (namely the attention sinks) may decrease the LA path’s contributions to the hybrid output, although, so far, while there is a notable difference in magnitude, it is not quite large enough to explain the full collapse of LA observed in LoLCATs. (2) We are also looking at differences in attention weights and their entropy, as well as the similarity of outputs between LA and softmax attention applied to the corresponding context. While we also observe an imbalance here, it is once again not large enough to explain the observed collapse. While, together, these analyses are likely to explain the observed performance gaps, individually they only appear to show part of the picture, and are therefore weak candidates for a robust unified metric.
> >
> > However,  we are happy to add such analysis to the appendix if the reviewer believes this would be a valuable addition. In the meantime, we have also clarified that we recommend quantifying the gaps in performance between hybrid and SWA-only, as well as no attention and LA-only, as the main thing to check for successful conversions of pre-trained transformers, and formalised this metric in section 5.
> >
> > If the reviewer feels that any further changes to the text are needed to better reflect the points above, we will be happy to accommodate.

---

### Official Review · Reviewer_Eoas · 2025-11-01

**Soundness:** 2
**Presentation:** 2
**Contribution:** 3
**Rating:** 4
**Confidence:** 4

**Summary:**

The paper identifies a previously overlooked failure mode in “hybrid” post-training recipes that convert pre-trained softmax Transformers into linear-attention (LA) / sliding-window-attention (SWA) hybrids: during light LoRA fine-tuning the model learns to ignore the LA branch and relies almost entirely on SWA, invalidating efficiency claims.

The authors provide (i) a reproducible diagnostic that ablates each branch at inference, (ii) an analysis tracing the collapse to the hybrid-output MSE transfer objective and to the small feature-map used by LoLCATs, and (iii) three mitigation strategies: (a) zero-shot inference-time ensembling, (b) HedgeCATs (attention-weight transfer + early-stopped hybrid LoRA), and (c) Scheduled Sliding-window Dropout (SSD).

Across Mistral-7B, Llama-3-8B and Llama-3.1-8B, the proposed recipes recover ≥95 % of base model accuracy on six zero-shot commonsense benchmarks while measurably increasing LA utilisation.

**Strengths:**

Originality
- First systematic evidence that hybrid conversion methods can silently collapse to SWA-only (Sec. 3.2, Table 1).
- Novel diagnostics (component ablation at inference) that can be immediately adopted by the community.

Quality & Rigor
- Extensive ablations isolating the transfer objective, φ dimension, and activation choice (Sec. 3.4, Tables 2–4).
- Three conceptually distinct fixes, each tested with the same protocol; best method (SSD) shows stable LA+SWA gap < 4 % over SWA-only (Table 10).

Clarity
- Reproducible experimental setup: frozen hyper-parameters, public checkpoints, LoRA details given (Sec. 3.1).
- Failure mode is intuitively explained and visually supported (Fig. 3, trend lines).

Significance
- Provides practical recipes that can be plugged into existing LoRA toolboxes with <1 % extra compute.

**Weaknesses:**

Limited Model & Task Diversity
- Only three instruction-tuned LLMs (7–8 B) and six short-context commonsense tasks; no evidence that collapse persists in larger models, in base (non-instruct) checkpoints, or in long-context evaluations (the motivating scenario, Sec. 1).
- Authors should add at least one 1–2 k token length benchmark (e.g., Scrolls QMSum) to validate efficiency claims under the intended use-case.

Missing Statistical Reliability
- All numbers appear to come from a single run; no standard errors or confidence intervals.
- Sec. 3.2 claims “SWA-only matches or slightly improves” but differences are ≤0.5 % on several tasks—insignificant without variance estimates.

Baseline Gaps
- Missing comparison with recent “full linear” SOTA HedgeHog using identical φ-size and LoRA budget (authors test HedgeHog only with weight-transfer, not the full pipeline).

Efficiency Claims Not Empirically Verified
- Paper asserts LA yields linear memory, but no wall-clock time, throughput, or peak-memory measurements are reported; without them the “computational efficiency” advantage (Abstract) is unvalidated.

Hyperparameter Sensitivity of SSD
- The Scheduled Sliding-window Dropout (SSD) is an elegant solution, but it introduces several new, sensitive hyperparameters, namely the dropout schedule and the window size schedule. The paper presents results for a few specific schedules (e.g., dropout of 0.9 -> 0.75 -> 0.5 in Fig 4a) but does not discuss how these schedules were chosen or how sensitive the model's performance is to them. This leaves open questions about the practicality and tuning cost of the method.

**Questions:**

Please see the questions in the weaknesses section above.

---

> ### Author Response · Authors · 2025-11-26
>
> Thank you very much for your constructive feedback. Please find our detailed answers below.
>
> > Only three instruction-tuned LLMs (7–8 B) and six short-context commonsense tasks; no evidence that collapse persists in larger models, in base (non-instruct) checkpoints, or in long-context evaluations (the motivating scenario, Sec. 1).
>
> >Authors should add at least one 1–2 k token length benchmark (e.g., Scrolls QMSum) to validate efficiency claims under the intended use-case.
>
> We thank the reviewer for these insightful suggestions. We have now repeated the experiments presented in Section 3.2 and confirm that the collapse persists across both smaller and larger model sizes with the addition of the Qwen model family (Appendix A.6.2), as well as in base checkpoints (Appendix A.6.3).
>
> We are still working on identifying an appropriate long-context benchmark. Although we have tested several options, including the reviewer’s suggestion, both SWA and LA, individually and combined, fail to recover any meaningful proportion of the base model’s performance in such a setting, irrespective of the conversion method used. While the reviewer is correct that long-context settings stand to benefit most from linear attention, to the best of our knowledge the field has yet to demonstrate an implementation of linear attention (without additional gating mechanisms) that achieves competitive performance in this regime. Because our work focuses specifically on the collapse of linear attention under hybrid conversion methods (using 1024-token sequences), existing long-context benchmarks may not be directly aligned with our setting. Our Discussion section mentions this limitation and suggests that a promising direction would be to expand conversion methods to gated formulations of linear attention.
>
> Nonetheless, we are continuing to run longer-context evaluations in the hope of identifying a benchmark that yields more informative results. We have also softened the wording in Section 1 to better reflect this gap between LA’s motivation and current performance.
>
> > All numbers appear to come from a single run; no standard errors or confidence intervals. Sec. 3.2 claims “SWA-only matches or slightly improves” but differences are ≤0.5 % on several tasks—insignificant without variance estimates.
>
> We do agree with the reviewer in that reporting results from a single run makes it difficult to distinguish signal from noise, especially when differences are very small. While we do control the initialisation of new components (appendix A.2.1), LoRA initialisation is still random. To this end, we have repeated the results presented in Table 1 with different LoRA initialisations to generate standard errors and have updated the table accordingly.
>
> On the other hand, we would like to clarify that the key result in the quoted statement “SWA-only […] matches or slightly improves average accuracy [of hybrid attention]” (Section 3.2) is precisely the fact that this difference is very small or even insignificant, as we seek to demonstrate that performance in hybrid attention can be almost entirely attributed to the fine-tuned SWA-only depending on the conversion method. However, to this end, we have updated the wording in the main text (Section 3.2) to reflect statistically significant gaps in performance, where appropriate.
>
> > Missing comparison with recent “full linear” SOTA HedgeHog using identical φ-size and LoRA budget (authors test HedgeHog only with weight-transfer, not the full pipeline).
>
> We thank the reviewer for this helpful observation. It made us realise that our description of HedgeHog relative to LoLCATs was misleading in the context of pre-trained conversions. While HedgeHog does employ full-parameter finetuning in some of its experiments, specifically training from scratch and task-specific conversions, it uses the same LoRA configuration as LoLCATs for pre-trained conversion experiments (e.g., Llama-2-7B in the original paper). Moreover, the SOTA claim derives from the more recent results reported in the LoLCATs paper, which also evaluate HedgeHog under these same LoRA settings.
>
> Accordingly, the results presented in Section 3.3, based on LoRA finetuning and the original φ-size, do correspond to the full HedgeHog pipeline, albeit finetuned on a different dataset. We have updated our descriptions of HedgeHog in section 3.3 to eliminate any ambiguity on this point.
>
> For completeness, we have additionally repeated the experiments in Section 3.3 using full-parameter finetuning (Appendix A.6.4). The results are consistent with their LoRA counterparts, though showing slightly lower average performance overall.

---

> > ### Author Response · Authors · 2025-11-26
> >
> > > Paper asserts LA yields linear memory, but no wall-clock time, throughput, or peak-memory measurements are reported; without them the “computational efficiency” advantage (Abstract) is unvalidated.
> >
> > We agree with the reviewer in that theoretical complexity does not always translate to wall-clock speedups, and that we need concrete measurements to substantiate the "efficiency" claims in the abstract. We have benchmarked efficiency measures of our hybrid models compared to the full softmax baseline and added a new table reporting these empirical metrics in the revised manuscript (section A.4).
> >
> > However, we find that both the theoretical complexity and empirical efficiency metrics have already been covered extensively in prior works [1-4] on linear and hybrid attention, including in the LoLCATs paper which we focus on replicating and improving. Furthermore, our current codebase does not seek to achieve the most efficient implementation of linear attention, which would likely require a kernelised implementation. Consequently, the measured wall-clock times will be misleading compared to the SWA or full attention components which use flash-attention. However, to provide a complete picture, we have included both the empirical efficiency table and a formal theoretical $\mathcal{O}(\cdot)$ complexity analysis (Appendix A.4).
> >
> > References
> >
> > [1] Michael Zhang, Simran Arora, Rahul Chalamala, Alan Wu, Benjamin Spector, Aaryan Singhal, Krithik Ramesh, and Christopher R´e. Lolcats: On low-rank linearizing of large language models. arXiv preprint arXiv:2410.10254,
> >
> > [2] Iz Beltagy, Matthew E Peters, and Arman Cohan. Longformer: The long-document transformer. arXiv preprint arXiv:2004.05150, 2020.
> >
> > [3] Simran Arora, Sabri Eyuboglu, Michael Zhang, Aman Timalsina, Silas Alberti, Dylan Zinsley,James Zou, Atri Rudra, and Christopher R ´e. Simple linear attention language models balance the recall-throughput tradeoff. arXiv preprint arXiv:2402.18668, 2024
> >
> > [4] Tsendsuren Munkhdalai, Manaal Faruqui, and Siddharth Gopal. Leave no context behind: Efficient infinite context transformers with infini-attention. arXiv preprint arXiv:2404.07143, 101, 2024.
> >
> > > The Scheduled Sliding-window Dropout (SSD) is an elegant solution, but it introduces several new, sensitive hyperparameters, namely the dropout schedule and the window size schedule. The paper presents results for a few specific schedules (e.g., dropout of 0.9 -> 0.75 -> 0.5 in Fig 4a) but does not discuss how these schedules were chosen or how sensitive the model's performance is to them. This leaves open questions about the practicality and tuning cost of the method.
> >
> > We thank the reviewer for pointing out this weakness. We agree that in order to make our method practical for other practitioners we should provide some guidance as to how the schedule hyperparameters should be decided. To this end, we have provided further motivation as per our choices in the experiments presented (Section 4.3, first paragraph). The key reason being that the model should initially have to rely more on LA outputs, then gradually gain access to more SWA context and outputs.
> >
> > In the meantime, we are ablating schedules. We will update the manuscript with our findings and include a dedicated section on hyperparameter recommendations.
> >
> > However, we are unlikely to be able to complete a full hyper-parameter search for all models before the end of the rebuttal period, given the large search space and our limited computational resources which are already quite busy with other additional experiments. To this end, as we fully agree with the reviewer, we explicitly mention that this hyper-parameter tuning is costly and present this is a weakness to the SSD method in the Discussion section (Limitations and Future Work paragraph).

---

### Author Response · Authors · 2025-11-21

Dear Area Chair and Reviewers,

We sincerely thank you for the insightful and constructive feedback. We are encouraged that the reviewers found our problem identification (the silent imbalance of SWA vs LA utilisation in hybrid linear conversions) to be novel, and the proposed core diagnostics and remedies to be sound and practical.

We acknowledge and appreciate the consistent feedback regarding the need for long-context validation, description of efficiency gains, further ablations, and quantification of LA utilisation in hybrid models. We are writing to confirm that we are actively running and finalising these experiments. We aim to upload a revised version of the manuscript early next week, alongside detailed responses to each of the points raised.

We would like to sincerely thank all of the reviewers for their suggestions, as we believe these additions will make for a much stronger paper.

Best regards,

The Authors

---

### Author Response · Authors · 2025-11-26
**General comment following initial rebuttals**

Dear Area Chair and Reviewers,

Following up on our previous message, we have completed the additional experiments requested during the review process and have uploaded a revised manuscript. We outline some key changes below but detail all answers and changes in individual comments to each review.

Summary of Major Updates:

1. Model Scaling & Generalisation (Appendix A.6.2): To address concerns regarding model diversity (Reviewer Eoas), we replicated our findings on the Qwen2.5 family (0.5B, 7B, and 14B). We confirm that the linear component collapse persists across these scales.
2. Base Model Evaluation (Appendix A.6.3): We verified our findings on base (non-instruction tuned) checkpoints for Mistral and Llama-3/3.1, confirming that the issue is fundamental to the conversion method and not an artifact of instruction tuning.
3. Efficiency Benchmarks (Appendix A.4, Table 5): Addressing Reviewer 8pxg and Eoas, we added empirical measurements comparing wall-clock time, samples per seconds and memory usage against FlashAttention-2 and Eager attention in long-context as well as theoretical complexity analysis.
4. Statistical Rigor: We re-ran our main ablations with multiple seeds and added standard errors to Table 1.
5. Window Size Ablation (Appendix A.6.1): We analysed the impact of varying the sliding-window size during training to further diagnose the mechanism of collapse.
6. Title Update: We have refined the paper title to "Untangling Component Imbalance in Hybrid Linear Attention Conversion Methods". We believe this more accurately reflects the core diagnostic contribution (identifying the SWA/Linear imbalance) that was highlighted as a key strength by Reviewers Eoas, nPh2, and 8pxg.

We believe these additions directly address the primary weaknesses raised regarding generalisation and efficiency validation. We have provided detailed responses to each reviewer below.

Best regards,

The Authors

---

### Author Response · Authors · 2025-12-03
**Final comments regarding long-context**

Dear Area Chair and Reviewers,

Multiple reviewers have requested long-context evaluations of our hybrid conversion methods. We respectfully maintain that such evaluations are not representative of our contributions and may be misleading in the context of this work.

Our paper focuses on hybrid conversion methods using standard linear attention (LA) in the short-context regime where such methods are originally validated. As we now document in Appendix A.3.1, linear attention architectures are fragile in long-context settings. Strong long-context performance typically requires: (1) explicit training close to target sequence lengths, (2) dedicated extrapolation mechanisms (e.g., RoPE scaling), or, most importantly, (3) architectural extensions (learned mixing terms, state decay, gated updates) - none of which are present in standard LA or the conversion methods we study. Even when LoLCATs and related approaches report long-context results, they require additional training at longer sequences. SUPRA achieves long-context performance through decay factors that effectively shorten the receptive field rather than demonstrating genuine long-context utilization. We have added this discussion to Appendix A.3.1 to provide comprehensive context on why long-context evaluation is not appropriate for our work.

Our work diagnoses and corrects the failure modes of LoLCAT-style conversions in their original short-context regime. Nevertheless, for completeness, we have conducted long-context evaluations (Appendix A.6.6). As anticipated, these results exhibit high variability and lack meaningful signal, as they operate in an under-specified regime without the additional training or architectural mechanisms that current work deems necessary for reliable long-context performance.

Our contributions lie in understanding and improving hybrid conversion methods in the short-context setting where they are designed to operate. We hope reviewers and ACs will evaluate our paper based on these core contributions rather than on experiments that fall outside our stated scope, architectural assumptions, and the validation regime of the methods we build upon.

We once again thank the reviewers for their thorough and constructive feedback across all aspects of our work. We believe the resulting revisions have substantially strengthened the paper, particularly in areas central to our contributions.

Best regards,

The Authors

---

### Meta-Review · Area_Chair_pYFh · 2025-12-28

**Summary:**

The authors provide insight into conversion strategies for linear attention schemes and propose several strategies to improve them. The initial reviews where negatively leaning (4,4,4,6), with several concerns, the most pressing ones about missing long-context evaluations and a tangible metric for measuring the impact of the linear attention branch. All in all, I think the authors did the best their could to respond to the given concerns. The answers are reasonable, make sense and are supported by evidence. Nevertheless, I do not believe the answers would have changed the reviewers opinion significantly. The main concerns about lacking applicability to long-context tasks (which are an important part of the original motivation) and questionable practical significance weigh too strong, even if all reviewers appreciate the insight provided by the author's analysis.

Since I believe this paper would have remained in a negatively-leaning or borderline state, I intend to follow the reviewers suggestion and recommend to reject the work in the current form. I encourage the authors to further think about how to more convincingly show the practical relevance, i.e. how to transfer the relevant practical insights gained into practice.

**Reviewer Concerns:**

*1) No long-context evaluations.* This is the main concern of multiple reviewers. Since the main motivation of improving the attention efficiency is to scale to large contexts, clear evidence of effectiveness in such settings would be desired. The authors where not able to provide such evidence. However, they spent considerable effort on trying and justifying the lack of such results. The main argument is that no fitting conversion method was shown to work in such a setting exists so that experiments cannot be easily transferred to such domain. They supported this statement by an explanation in the appendix, going through all relevant recent work, and experiments that show that reasonable results cannot be obtained on long-context tasks. In general, while I believe the authors did the best they could do support their stance, this still raises questions about the practical relevance of the proposed method, if it cannot be applied in situations it is targeting in high-level motivation.

*2) Limited Model diversity.* The authors addressed this by providing additional experiments on Qwen2.5.

*3) Missing Statistical Reliability.* The authors addressed this by providing standard deviations for a subset of experiments.

*4)  Missing comparison to “full linear” SOTA HedgeHog .* Successfully rebutted and addressed.

*5) Efficiency claims not validated.* The authors addressed this by providing wall clock times, memory consumption, and a theoretical runtime analysis. It remains questionable if conversion into linearized attention can become competitive to flash attention, especially if it cannot be applied to long-context domains.

*6) Hyperparameter Sensitivity.* The authors addressed this by adding more discussions and recipes for choosing hyperparameters.

*7) Additional Diagnosis metric for contribution of linear attention.* The authors discussed several options and reported that they did not find a meaningful metric. If the main contribution is the insight of which strategies serve best to enable linear attention conversion, such a metric would strengthen the claims made in the work.

**Reviewer Scores:**

Reviewer ry4B would not have increased as indicated by a discussion post. The main concerns of no supporting results on long-context tasks as well as no tangible metric for the use of the linear branch weigh too strong and question generality and practical applicability. My guess is that Reviewer nPh2 would have a similar opinion, as the main concerns are similar. Only Reviewer Eoas might have increased the score, given that several of the concerns were addressed. However, similarly to the previous two, concerns about long-context experiments were present and not fully addressed.

All in all, this would leave the paper in (4,4,6,6) or (4,4,4,6) state of scores, post-discussion.

---

### Decision · Program_Chairs · 2026-01-26

Reject